# One Wave To Explain Them All: A Unifying Perspective On Feature Attribution

**Gabriel Kasmi** [1 2]  **Amandine Brunetto** [1]  **Thomas Fel** [3]  **Jayneel Parekh** [4]

## Abstract

Feature attribution methods aim to improve the transparency of deep neural networks by identifying the input features that influence a model's decision. Pixel-based heatmaps have become the standard for attributing features to high-dimensional inputs, such as images, audio representations, and volumes. While intuitive and convenient, these pixel-based attributions fail to capture the underlying structure of the data. Moreover, the choice of domain for computing attributions has often been overlooked. This work demonstrates that the wavelet domain allows for informative and meaningful attributions. It handles any input dimension and offers a unified approach to feature attribution. Our method, the **W**avelet **A**ttribution **M**ethod (WAM), leverages the spatial and scale-localized properties of wavelet coefficients to provide explanations that capture both the *where* and *what* of a model's decision-making process. We show that WAM quantitatively matches or outperforms existing gradient-based methods across multiple modalities, including audio, images, and volumes. Additionally, we discuss how WAM bridges attribution with broader aspects of model robustness and transparency. Project page: https://gabrielkasmi.github.io/wam/.

## 1. Introduction

Deep neural networks are increasingly being deployed in various applications. However, their opacity poses a significant challenge, especially in safety-critical domains, where understanding the decision-making process is crucial, for instance, in tumor detection (Pooch et al., 2020) or obstacle identification (Sun et al., 2022).

[1]Mines Paris - PSL University, Paris, France [2]RTE France, Paris La Défense, France [3]Kempner Institute, Harvard University, Cambridge, MA, United-States [4]ISIR, Sorbonne Université, Paris, France. Correspondence to: Gabriel Kasmi <gabriel.kasmi[at]minesparis.psl.eu>.

*Proceedings of the 42nd International Conference on Machine Learning*, Vancouver, Canada. PMLR 267, 2025. Copyright 2025 by the author(s).

This opacity motivated the rise of Explainable Artificial Intelligence (XAI) techniques to provide human-understandable explanations of model decisions. While XAI has been predominantly applied in image classification, it is also extending into other fields, such as audio and volumes classification (Parekh, 2023; Paissan et al., 2024; Chen et al., 2021; Zheng et al., 2019). Currently, the most popular XAI methods are feature attribution methods (Shrikumar et al., 2017; Sundararajan et al., 2017; Smilkov et al., 2017; Fel et al., 2021; Novello et al., 2022; Muzellec et al., 2024) which consist in generating saliency maps, i.e., heatmaps highlighting the important pixels on the input image (Zeiler & Fergus, 2014b). The same principle has been expanded for audio and volumes, where most feature attributions are computed on 2D projections of the input modality. However, the pixel domain does not inherently capture the structural properties of signals. So far, existing works have overlooked the question of the domain in which feature attribution is computed. However, signal processing theory states that meaningful features often exist at multiple scales and are better represented in transformed domains that account for spatial and frequency information. This underlines the need for an alternative representations that preserves these properties to provide more faithful explanations.

Wavelets offer a hierarchical decomposition that retains both spatial and frequency information, unlike pixel-based methods that lose structural context. This makes wavelets a stronger foundation for interpreting model decisions across diverse modalities (or signals), as wavelets are inherently low-level features defined across various signal dimensions. This work introduces the **W**avelet **A**ttribution **M**ethod (WAM), which leverages wavelet coefficients as the basis features for computing attributions. Therefore, WAM provides a unification of the feature attribution domain on components (wavelet coefficients) that are interpretable as low-level features of the input signal. We expand popular feature attribution methods, namely SmoothGrad (Smilkov et al., 2017) and Integrated Gradients (Sundararajan et al., 2017), within the wavelet domain, thus providing a generalization of feature attribution to any square-integrable modality. WAM preserves the original structure of the data without converting it into a 2D representation.

Figure 1 illustrates our approach, which computes the gradient of a classification model with respect to the wavelet co-

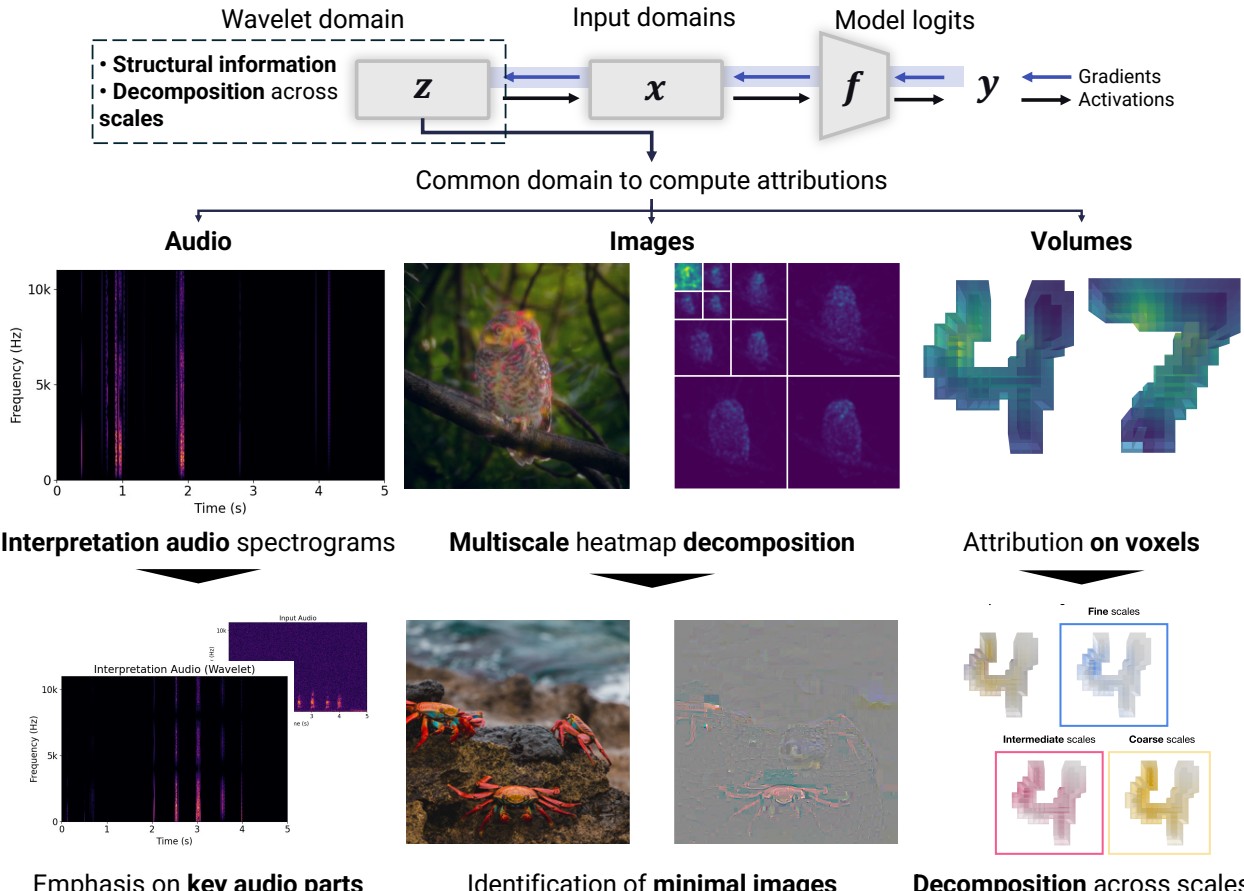

*Figure 1.* **Explaining any input modality by decomposing the model's decision in the wavelet domain.** WAM computes the gradient of the model's prediction with respect to the wavelet coefficients of the input modality (audio, image, volume). Unlike pixels, wavelet coefficients preserve structural information about the input signal, offering deeper insights into the model's behavior and going beyond *where* it focuses.

efficients of the input signal. These coefficients are defined for any square-integrable function, regardless of dimension, and are localized in both space and scale, allowing them to capture information across different scales. For images and volumes, wavelet coefficients capture shapes, edges, and textures, while for audio signals, they represent transient patterns. Consequently, feature attribution in the wavelet domain enables us to interpret audio, highlight important areas and patterns in images, or identify key regions in volumes, all in a post-hoc manner, i.e., using only a trained model and a single backward pass.

We demonstrate the superiority of WAM through extensive empirical evaluations across a diverse set of metrics, model architectures, and datasets, showing its robustness and versatility. We also discuss the novel insights and connections that our method permits, such as connecting feature attribution and the characterization of a model's robustness

or filtering the important part of an audio signal without requiring a to train an explanation method.

In summary, the following are the key contributions of this work:

- We introduce WAM, a novel feature attribution method that operates in the wavelet domain, ensuring that the attributions align with the intrinsic properties of the data, whether it be audio, images, or volumes.

- We demonstrate the effectiveness of WAM through extensive empirical evaluations across diverse models, datasets, and performance metrics.

- We illustrate the relevance of transitioning to the wavelet domain for feature attribution by showcasing how WAM offers novel insights into model transparency.

## 2. Related works

### 2.1. Feature attribution methods

**Images.** Computer vision has supported the development of numerous post-hoc feature attribution methods (Baehrens et al., 2010). These techniques are applied to a trained model and estimate the importance of each pixel or region of an image based on its contribution to the model's prediction, i.e., estimate a saliency map. Post-hoc methods either leverage internal model information, such as gradients (Baehrens et al., 2010; Simonyan et al., 2014; Zeiler & Fergus, 2014b; Springenberg et al., 2014; Sundararajan et al., 2017; Smilkov et al., 2017; Muzellec et al., 2024; Han et al., 2024; Binder et al., 2016) or apply perturbations to the input space (Lundberg & Lee, 2017; Ribeiro et al., 2016; Petsiuk et al., 2018; Fel et al., 2021; Novello et al., 2022; Kasmi et al., 2023a).

**Audio.** The methods introduced in computer vision have been extended to audio classification in three directions. The first one explores the use of saliency methods to highlight key features for audio classifiers processing spectrograms (Becker et al., 2024; Won et al., 2019) or waveforms (Muckenhirn et al., 2019). The second direction involves variants of LIME (Ribeiro et al., 2016) algorithm, proposed for different types of audio classification tasks (Mishra et al., 2017; 2020; Haunschmid et al., 2020; Chowdhury et al., 2021; Wullenweber et al., 2022). The third pursues the development of methods to generate listenable interpretations for audio classifiers by leveraging the hidden representations (Parekh et al., 2022; Paissan et al., 2024).

**Volumes.** 3D data can be represented as point clouds or voxels. Point clouds offer an exact representation of the data but are unstructured. Voxels, conversely, are a discretized but structured representation of the data, making them suitable for processing with techniques such as 3D convolutions. Most explainability techniques for 3D data focused on explaining point clouds. Chen et al. (2021); Schinagl et al. (2022); Gupta et al. (2020); Zheng et al. (2019) introduced techniques to generate visual explanations for interpretability of 3D object detection and classification networks. They highlight critical features in point cloud by adapting 2D image-based saliency techniques (Gupta et al., 2020; Zheng et al., 2019), by using a perturbation-based approach (Schinagl et al., 2022) or by proposing a 3D variant of LIME (Tan & Kotthaus, 2022). Explainability on volumes remains limited. A few works (Yang et al., 2018; Mamalakis et al., 2023; Gotkowski et al., 2021) have proposed attention maps on 2D slices of 3D medical scans using 3D-GradCAM.

Current feature attribution methods rely on a feature space that overlooks the inherent temporal, spatial, or geometric relationships within the data. Even worse, projecting explanations into the two-dimensional pixel domain for 1D audio or 3D volumes further distorts these relationships, resulting in explanations that fail to capture the full complexity of the model's decision-making process, ultimately reducing the relevance of the attributions.

### 2.2. Wavelet-based explainability

From an explainability perspective, we argue that the wavelet domain is more informative for feature attribution than the pixel domain, as it enables the extraction of interpretable low-level features from input signals. In particular, the dyadic wavelet decomposition of a signal isolates distinct patterns within the input signal. In images, wavelet coefficients capture edges, textures, and patterns at different scales and orientations, aligning with human perception. For instance, wavelets coincide in some cases with Gabor filters (Gabor, 1946), which have been widely used for pattern analysis. In audio, they separate transient components from slowly varying structures, facilitating the identification of speech formats or sudden bursts of sound.

Surprisingly, hardly a handful of works explored the use of wavelets for post-hoc explainability. CartoonX (Kolek et al., 2022) and ShearletX (Kolek et al., 2023) extended the Meaningful Perturbation framework (Fong & Vedaldi, 2017) into the wavelet and shearlet domains, respectively, allowing to highlight not just salient pixel regions but also relevant textures and edge structures. Recently, the Wavelet Scale Attribution Method (WCAM, Kasmi et al., 2023a), transposed Sobol-based feature attribution (Fel et al., 2021) to the wavelet domain, demonstrating the feasibility of multi-scale attributions for images. Wavelet coefficients have been used to construct scattering transforms (Bruna & Mallat, 2013; Andén & Mallat, 2014), i.e., fixed feature extractors that create a translation-invariant representation of an input, stable to deformations. This aligns with the properties of trained feature extractors like convolutional neural networks (CNNs) without any training. This model and its descendants, such as the scattering spectra (Cheng et al., 2024), can be considered intrinsically interpretable models (Flora et al., 2022) and have been applied in fields where model understanding is crucial, such as finance (Leonarduzzi et al., 2019) and astrophysics (Cheng & Ménard, 2021).

Finally, Ha et al. (2021) proposed a method which aims to distill information from a trained neural network into a wavelet transform, resulting in a highlight predictive, concise and computationally efficient model, whose properties (e.g., multi-scale structure) make it easy to interpret. This model was applied to cosmological parameter inference and molecular-partner prediction.

**a) Original image**

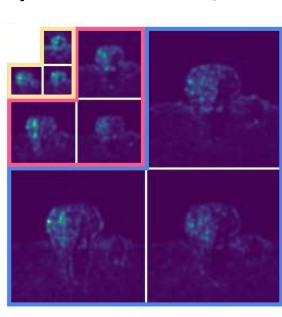

**b) Wavelet heatmap**

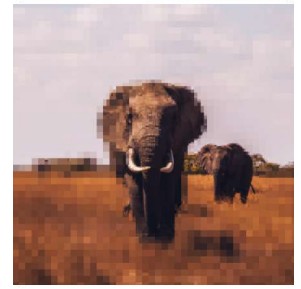

**c) Image reconstruction**

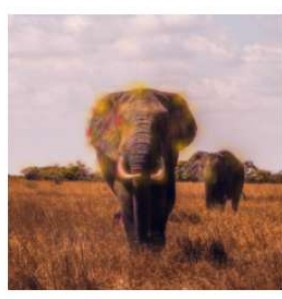

**No details** needed in the **background**

**High-resolution** detail is essential in the **center area**

**d) Heatmap and decomposition across scales**

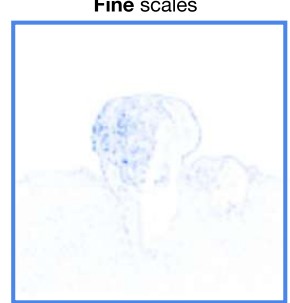

Fine scales        Intermediate scales        Coarse scales

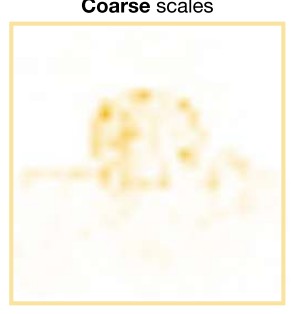

*Figure 2.* **WAM for images.** Our method decomposes important components at multiple scales, revealing *what* the model focuses on. The same spatial location encodes different structural elements at different scales: the model predicts the elephant because of its trunk and ears but needs both the fine texture details (blue), the intermediate-scale contours (red), and the coarse edges (yellow). Such a decomposition is not possible with traditional feature attribution in the pixel domain, and this hierarchical decomposition provides an interpretable view of model attributions.

## 3. Feature attribution in the wavelet domain

**Notations.** Throughout, we let $\mathcal{X} = (\Omega, \mathcal{F}, \mu)$ be a measure space with set $\Omega$, $\sigma$-algebra $\mathcal{F}$, and measure $\mu$. $\mathcal{H} = L^2(\mathcal{X}, \mu)$ denotes the Hilbert space of square-integrable functions on $\mathcal{X}$. Let $\boldsymbol{f} \in \mathcal{H}$ represent a predictor function (e.g., a classifier), which maps an input $\boldsymbol{x} \in \mathcal{X}$ to an output $\boldsymbol{f}(\boldsymbol{x}) \in \mathcal{Y}$. We denote $\boldsymbol{g} \in \mathcal{H}$ a square-integrable function.

### 3.1. Multiscale decompositions

**Overview.** Wavelets decompose signals into components that retain both spatial and frequency information, unlike Fourier transforms, which capture only frequency and discard spatial or temporal context. The choice of the wavelet determines what kind of structural components will be highlighted on the image. Its choice is therefore grounded in the application at hand. We refer the reader to Appendix A.2 for a discussion of the choice of the mother wavelet and its impact on the interpretation of the WAM.

**Definition.** A *wavelet* is an integrable function $\psi \in \mathcal{H} = L^2(\mathbb{R}^n)$ that is normalized, centered at the origin, and has zero mean (i.e., $\int_{\mathbb{R}^n} \psi(\boldsymbol{x}) \, d\boldsymbol{x} = 0$). In contrast to sine waves, which are localized in frequency but not in space, wavelets are localized in both *space* and *frequency* domains. To analyze a function or a signal $\boldsymbol{g} \in \mathcal{H}$, we define a family of functions $\mathcal{D}$ obtained by dilating and translating a mother wavelet $\psi$:

$$\mathcal{D} = \left\{ \boldsymbol{\psi}_{\boldsymbol{\lambda}, \boldsymbol{b}}(\boldsymbol{x}) = \frac{1}{\boldsymbol{\lambda}^{n/2}} \psi \left( \frac{\boldsymbol{x} - \boldsymbol{b}}{\boldsymbol{\lambda}} \right) \right\}_{\boldsymbol{b} \in \mathbb{R}^n, \, \boldsymbol{\lambda} > 0}, \quad (1)$$

where $\boldsymbol{b}$ is a translation and $\boldsymbol{\lambda}$ is a scale factor. This family gives rise to the continuous wavelet transform (CWT) of $\boldsymbol{g}$, defined as

$$\mathcal{W}(\boldsymbol{g})(\boldsymbol{\lambda}, \boldsymbol{b}) = \langle \boldsymbol{g}, \boldsymbol{\psi}_{\boldsymbol{\lambda}, \boldsymbol{b}} \rangle$$
$$= \int_{\mathbb{R}^n} \boldsymbol{g}(\boldsymbol{x}) \frac{1}{\boldsymbol{\lambda}^{n/2}} \overline{\psi} \left( \frac{\boldsymbol{x} - \boldsymbol{b}}{\boldsymbol{\lambda}} \right) d\boldsymbol{x}. \quad (2)$$

This operation can be interpreted as a convolution with a time-reversed and dilated version of the complex-conjugate $\overline{\psi}$ of $\psi$ (i.e., a linear filtering operation), and thus defines a linear operator on $\mathcal{H}$ (Mallat, 2008).

**Discrete Wavelet Transform.** When the scale parameter is restricted to dyadic values $\boldsymbol{\Lambda} = \{2^{-j} : j \in \mathbb{Z}\}$ and

translations lie on a discrete grid, the continuous family becomes a countable set. Under appropriate admissibility and regularity conditions on $\psi$, this discrete family forms an orthonormal basis or a tight frame for $\mathcal{H}$, giving rise to the *Discrete Wavelet Transform* (DWT).

The DWT is defined by projecting a signal $\boldsymbol{g} \in L^2(\mathbb{R}^n)$ onto a dyadic family of wavelets obtained by discretizing the scale and translation parameters. For dyadic scales $\boldsymbol{\lambda} = 2^{-j}$ and integer translations $\boldsymbol{b} = 2^{-j}\mathbf{k}$, we define the family

$$\left\{ \boldsymbol{\psi}_{j,\mathbf{k}}(\boldsymbol{x}) = 2^{jn/2}\boldsymbol{\psi}(2^j\boldsymbol{x} - \mathbf{k}) \right\}_{\mathbf{k} \in \mathbb{Z}^n, \, j \in \mathbb{Z}}. \quad (3)$$

The DWT coefficients are given by the inner products

$$\mathcal{W}_{\mathrm{DWT}}(\boldsymbol{g})(j, \mathbf{k}) = \langle \boldsymbol{g}, \boldsymbol{\psi}_{j,\mathbf{k}} \rangle = \int_{\mathbb{R}^n} \boldsymbol{g}(\boldsymbol{x})\overline{\boldsymbol{\psi}_{j,\mathbf{k}}(\boldsymbol{x})} \, \mathrm{d}\boldsymbol{x}. \quad (4)$$

These coefficients represent the content of $\boldsymbol{g}$ at scale $2^{-j}$ and position $2^{-j}\mathbf{k}$. Under the appropriate admissibility and regularity conditions on $\psi$, the DWT is an invertible linear transform: the signal $\boldsymbol{g}$ can be exactly reconstructed from its wavelet coefficients $\mathcal{W}_{\mathrm{DWT}}(\boldsymbol{g})(j, \mathbf{k})$ via the inverse transform

$$\forall \boldsymbol{x} \in \mathbb{R}^n, \quad \boldsymbol{g}(\boldsymbol{x}) = \sum_{\substack{j \in \mathbb{Z} \\ \mathbf{k} \in \mathbb{Z}^n}} \langle \boldsymbol{g}, \boldsymbol{\psi}_{j,\mathbf{k}} \rangle \cdot \boldsymbol{\psi}_{j,\mathbf{k}}(\boldsymbol{x}). \quad (5)$$

Mallat (1989) showed that the dyadic wavelet transform of a signal $\boldsymbol{g}$ can be computed by applying a high-pass filter $h$, followed by downsampling by a factor of two, to obtain the *detail* coefficients, and similarly applying a low-pass filter $g$, followed by downsampling, to obtain the *approximation* coefficients. Iteratively applying this to the approximation coefficients yields a multilevel transform, where the $j^{\mathrm{th}}$ level captures information at scales corresponding to frequency bands between $2^j$ and $2^{j-1}$ octaves. For input signals $\boldsymbol{g}$ of dimension greater than one, the detail coefficients can be decomposed into directional components. In the 2D case (e.g., images), these typically correspond to vertical, horizontal, and diagonal orientations.

### 3.2. Feature attribution in the wavelet domain

**Problem formalization.** Let $\boldsymbol{f}$ be a classifier and $\boldsymbol{x}$ an input (e.g., an image, an audio, or a volume). The classifier $\boldsymbol{f}$ maps the input to a class $c$ as $\boldsymbol{y}_c = \arg\max_{c \in \mathcal{C}} \boldsymbol{f}(\boldsymbol{x}) \equiv \boldsymbol{f}_c(\boldsymbol{x})$ with a slight abuse of notation. We recall that the original saliency map of the classifier $\boldsymbol{f}$ for class $c$ is then given by $\gamma_{\mathrm{Sa}}(\boldsymbol{x}) = |\nabla_{\boldsymbol{x}} \boldsymbol{f}_c(\boldsymbol{x})|$ where $c$ denotes the class of interest. The saliency map is defined under the condition that the $\boldsymbol{f}_c$'s are piecewise differentiable (Simonyan et al., 2014). It highlights the most influential (in terms of the absolute value

of the gradient) components in the input $\boldsymbol{x}$ for determining the model's $\boldsymbol{f}$ decision. The higher the value, the greater the importance of the corresponding region.

However, moving from one pixel to the next is only a shift in the spatial domain and does not capture relationships between scales or frequencies. Therefore, we argue that the pixel domain is not well suited for explaining *what* the model is seeing on the image. On the other hand, the wavelet decomposition of an image – and more broadly of any differentiable modality – provides information on the structural components of the modality. Therefore, computing the gradient of $\boldsymbol{f}$ with respect to the wavelet transform of $\boldsymbol{x}$ will enable us to understand the model's reliance on features such as textures, edges, or shapes in the case of images and volumes and transients, or harmonics in sounds.

Denoting $\boldsymbol{z} = \mathcal{W}(\boldsymbol{x})$ the discrete wavelet transform of $\boldsymbol{x}$, since $\mathcal{W}$ is invertible, we can define the saliency map in the wavelet domain as

$$\gamma_{\mathrm{Sa}}(\boldsymbol{z}) = \left| \frac{\partial \boldsymbol{f}_c(\boldsymbol{x})}{\partial \boldsymbol{z}} \right| = \left| \frac{\partial \boldsymbol{f}_c(\boldsymbol{x})}{\partial \boldsymbol{x}} \cdot \frac{\partial \mathcal{W}^{-1}(\boldsymbol{z})}{\partial \boldsymbol{z}} \right|, \quad (6)$$

using the fact that $\boldsymbol{x} = \mathcal{W}^{-1}(\boldsymbol{z})$ and where $\dfrac{\partial \boldsymbol{f}_c(\boldsymbol{x})}{\partial \boldsymbol{x}}$ denotes the gradient of the classifier output with respect to the input image and $\dfrac{\mathcal{W}^{-1}(\boldsymbol{z})}{\partial \boldsymbol{z}}$ is the Jacobian matrix of the inverse wavelet transform. In practice, to retrieve Equation (6), we require the gradients on $\mathcal{W}(\boldsymbol{x})$ and directly evaluate $\dfrac{\partial \boldsymbol{f}_c(\mathcal{W}^{-1}(\boldsymbol{z}))}{\partial \boldsymbol{z}}$.

A remarkable property of this framework is that it **accommodates any input dimension**, and thus is **modality-agnostic**. Therefore, we can apply it – and leverage its properties – to numerical signals such as audio (1D signals), images (2D signals), or volumes (3D signals). In this paper, we demonstrate the superiority of this method in the 1D, 2D and 3D settings compared to other domain-specific methods.

**Smoothing.** Saliency maps computed following Equation (6) can fluctuate sharply at small scales as $\boldsymbol{f}_c$ is not continuously differentiable. To yield smoother explanations, smoothing consists in averaging the explanation over a set of noisy samples (Smilkov et al., 2017). Analogously, we propose to calculate

$$\gamma_{\mathrm{SG}}(\boldsymbol{z}) = \frac{1}{n} \sum_{i=1}^{n} \nabla_{\tilde{\boldsymbol{z}}} \boldsymbol{f}(\mathcal{W}^{-1}(\tilde{\boldsymbol{z}})), \quad (7)$$

where $\tilde{\boldsymbol{z}} = \mathcal{W}(\boldsymbol{x} + \boldsymbol{\delta})$ and $\boldsymbol{\delta} \sim \mathcal{N}(0, I\sigma^2)$. The number of samples, $n$, needed to compute the approximation of the smoothed gradient and the standard deviation $\sigma^2$ are hyperparameters. To transpose this method to the wavelet domain, we add noise to the input before computing its wavelet transform. We refer to this method as $\mathrm{WAM}_{SG}$ throughout the

rest of the paper. In Appendix A.1, we illustrate the enhancement of the quality of the explanation after applying the smoothing to the gradients as described in Equation (7).

**Path integration.** Another approach to derive smooth explanations from the model's gradients consists in averaging the gradient values along the path from a baseline state to the current value. The baseline state is often set to zero, representing the complete absence of features. This technique, Integrated Gradients (Sundararajan et al., 2017), satisfies two axioms, *sensitivity* and *implementation invariance*. Sensitivity states that "*for every input and baseline that differ in one feature but have different predictions, then the differing feature should be given a non-zero attribution*" and implementation invariance that "*the attributions are always identical for two functionally equivalent networks*". We adapt the Integrated Gradient method from the image domain to the wavelet domain. Denoting $z = \mathcal{W}(x)$, we evaluate

$$\gamma_{\text{IG}} = (z - z_0) \cdot$$
$$\int_0^1 \frac{\partial f_c \left( \mathcal{W}^{-1} \left( z_0 + \alpha(z - z_0) \right) \right)}{\partial z} \, d\alpha, \quad (8)$$

where $z_0$ denotes the baseline state of the wavelet decomposition of $x$. We refer to this implementation of WAM as $\text{WAM}_{IG}$.

Appendix A.1 illustrates the enhancement of the quality of the explanation after applying the smoothing to the gradients as described in Equation (7) and after integrating the gradients, as described in Equation (8).

**Theoretical properties.** WAM extends feature attribution into the wavelet domain. The main focus of the paper is the proposal and the practical usage of WAM, however, it appears that $\text{WAM}_{IG}$ inherits from the theoretical properties of the vanilla integrated gradients method, and especially sensitivity. We refer the reader to Appendix B for a discussion on the theoretical aspects of WAM.

## 4. Evaluation and applications

### 4.1. Quantitative evaluation

**Evaluation metrics.** We evaluate WAM against gradient-based methods, as they are more reliable and efficient than perturbation-based methods (Crabbé & van der Schaar, 2023). We evaluate the **Faithfulness**, which we define as the difference between the **Insertion** and **Deletion** scores, thereby following the definition proposed by Muzellec et al. (2024). We refer the reader to Chan et al. (2022) for a discussion of the alternative definitions of Faithfulness. Faithfulness is effective in evaluating attribution methods (Samek

et al., 2016; Li et al., 2022). Insertion and deletion (Petsiuk et al., 2018) have been widely used in XAI to evaluate the quality of feature attribution methods (Fong & Vedaldi, 2017). The deletion measures the evolution of the prediction probability when one incrementally removes features by replacing them with a baseline value according to their attribution score. Insertion consists in gradually inserting features into a baseline input.

Given a model $f$ and an explanation functional $\gamma$, the Faithfulness $F$ is given by

$$F(f, \gamma) = \text{Ins}(f, \gamma) - \text{Del}(f, \gamma). \quad (9)$$

We provide a detailed derivation of the Insertion and the Deletion scores in Appendix C.2. They were initially defined in the context of images, but we propose to expand them to audio and volumes.

**Evaluation setting.** To the best of our knowledge, no prior work has evaluated a feature attribution method across multiple input modalities. To address this gap, we designed a comprehensive evaluation framework spanning diverse datasets: ESC-50 (Piczak, 2015) for audio, ImageNet (Russakovsky et al., 2015) for images, and MedMNIST3D (Yang et al., 2023) for volumes. For each modality, we picked an arbitrary model and applied the same feature attribution methods to ensure consistency. We considered four feature attribution methods: SmoothGrad (Smilkov et al., 2017), Saliency (Simonyan et al., 2014), Integrated Gradients (Sundararajan et al., 2017) and GradCAM (Selvaraju et al., 2017). We use the Python library Captum (Kokhlikyan et al., 2020) for consistently implementing existing methods on our datasets for audio and images. For volumes, we integrated WAM as an additional method within the evaluation framework LATEC (Klein et al., 2024).

**Results.** Table 1 presents the performance of WAM. Our method consistently outperforms traditional attribution methods, especially for images and volumes, while achieving competitive results for audio. We can see that $\text{WAM}_{IG}$ outperforms $\text{WAM}_{SG}$. As further discussed in Appendix A.1, this improved performance arises because path integration captures inter-scale dependencies, revealing the relative importance of each scale in the model's prediction. Overall, these results highlight the relevance of using wavelets, rather than pixels, as the basis for feature attribution.

In Appendix C.3, we evaluate WAM using additional datasets, metrics, and alternative model topologies, and compare its accuracy with more methods across all three input modalities. Our results further emphasize that WAM consistently matches or surpasses the performance of existing methods.

*Table 1.* **Faithfulness (Faith), Insertion (Ins) and Deletion (Del) scores across modalities.** Evaluations are conducted on 400 samples from ESC-50 (fold 1), 1,000 images from ImageNet's validation set and the full AdrenalMNIST3D test set (298 samples). WAM outperforms all other methods on image and volume modalities for both Insertion and Deletion metrics, meaning that it is good at identifying which features influence the most the model's decision. For audio, it achieves state-of-the-art performance in Insertion and performs as well as other methods for Deletion and Faithfulness. Best results are **bolded** and second best underlined.

| | **Audio** | | | **Images** | | | **Volumes** | | |
|---|---|---|---|---|---|---|---|---|---|
| *Model* | *ResNet* | | | *EfficientNet* | | | *3D Former* | | |
| *Dataset* | *ESC-50* | | | *ImageNet* | | | *AdrenalMNIST3D* | | |
| | Ins (↑) | Del (↓) | Faith (↑) | Ins(↑) | Del (↓) | Faith (↑) | Ins (↑) | Del (↓) | Faith (↑) |
| Integrated Gradients | 0.267 | **0.047** | **0.264** | 0.113 | 0.113 | 0.000 | 0.666 | 0.743 | -0.077 |
| SmoothGrad | 0.251 | 0.067 | 0.184 | 0.129 | 0.119 | 0.010 | 0.680 | 0.731 | -0.051 |
| GradCAM | 0.274 | 0.201 | 0.072 | 0.364 | 0.303 | 0.061 | 0.689 | 0.744 | -0.055 |
| Saliency | 0.220 | 0.154 | 0.066 | 0.148 | 0.140 | 0.008 | 0.751 | 0.742 | 0.009 |
| $\text{WAM}_{IG}$ (ours) | 0.436 | 0.260 | 0.176 | **0.447** | **0.049** | **0.370** | **0.719** | **0.621** | **0.098** |
| $\text{WAM}_{SG}$ (ours) | **0.449** | 0.252 | 0.197 | 0.419 | 0.097 | 0.350 | 0.718 | 0.648 | 0.070 |

## 4.2. Exploring how WAM renews feature attribution

Beyond its strong quantitative performance, WAM also offers new qualitative insights by disentangling features at different scales. This section highlights three representative use cases. We provide more examples in Appendix D.

**Images: bridging attribution and robustness.** In computer vision, several works document a correlation between the reliance on low frequency components of the image and the robustness of the model to adversarial or natural corruptions (Zhang et al., 2022; Chen et al., 2022; Wang et al., 2020). This property is grounded in the frequency or spectral bias of models (Rahaman et al., 2019) and has been highlighted by filtering out frequency bands from the Fourier transform of the input images. As the Fourier magnitude discards spatial information, frequencies are removed in all the image.

WAM quantifies the importance of each scale in the final prediction. Scales in the wavelet domain correspond to dyadic frequency ranges in the Fourier domain. Therefore, by summing the importance of wavelet coefficients at a given scale, we can deduce the importance of the corresponding frequency range in the model's prediction. This estimation can be straightforwardly derived from the explanation and does not require multiple perturbations as done in previous works (Zhang et al., 2022; Chen et al., 2022). As a result, WAM can be used to evaluate the robustness of a model.

To show this property, we compare a set of ResNet models. The baseline is trained using the standard empirical risk minimization (Vanilla ResNet). The others are trained using three adversarial training methods: ADV (Madry et al.,

2018), ADV-Fast (Wong et al., 2020), ADV-Free (Shafahi et al., 2019).

Figure 3 averages the importance of each scale for 1,000 explanations computed from ImageNet. For each image, we normalize the importance to highlight the relative reliance on different scales. As expected, the vanilla ResNet relies more on fine scales (high frequencies) than the adversarially trained models. WAM consistently retrieves existing results from the robustness literature while avoiding the need for complex experimental settings, thus paving the way for easier assessment of the robustness of a model.

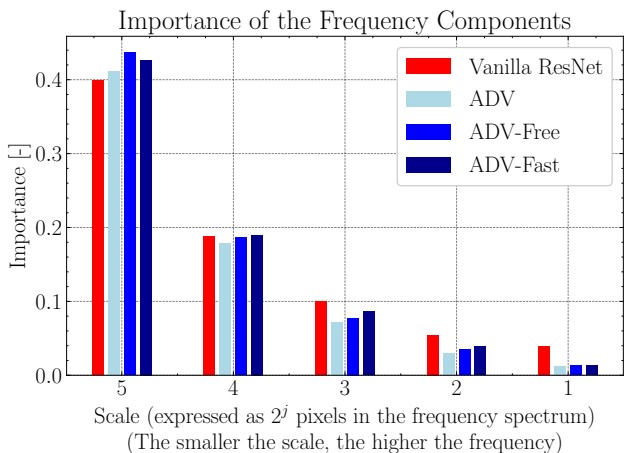

*Figure 3.* **Model robustness assessment with $\text{WAM}_{IG}$.** Each bar indicates the importance of each scale in the model's prediction. Importances are averaged over 1,000 explanations computed from ImageNet and normalized. The adversarially robust models rely more on coarse-scale (low frequency) features than the vanilla ResNet, showing that WAM recovers results from the literature.

**Volumes: revealing multi-scale structures.** We retrieve on voxels the same decomposition as for images or audio. Figure 4 highlights the significance of the edges at larger scales, whereas, at smaller scales, the importance becomes increasingly concentrated at the center of the digit. This multi-scale decomposition is particularly valuable for volumetric data, as it allows us to disentangle fine-grained details from broader structural patterns, which is crucial in many applications. For instance, in medical imaging, larger-scale features may correspond to organ boundaries or lesion contours, while finer scales capture subtle textural variations indicative of disease. Similarly, in 3D object recognition, coarse scales help identify overall shapes, whereas finer scales refine the details essential for distinguishing between similar objects. To the best of our knowledge, WAM is the first method to demonstrate such a decomposition on shapes, offering a new perspective on how models process hierarchical spatial information in 3D data.

**Heatmap and disentanglement across scales**

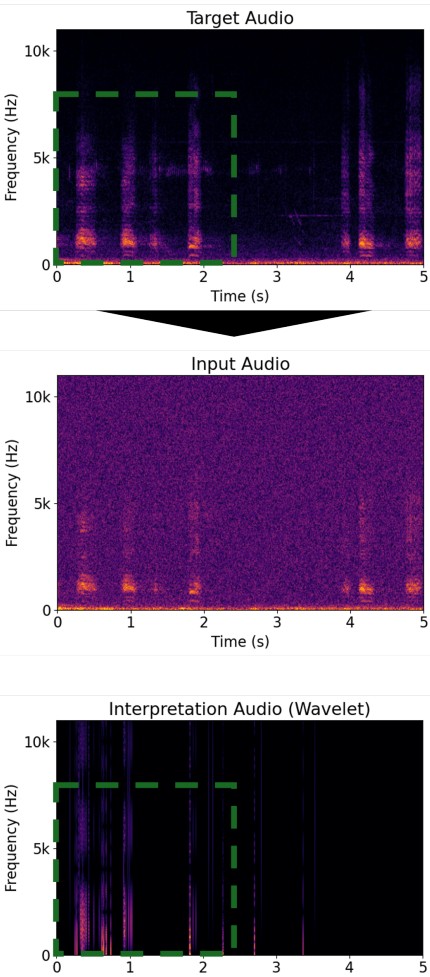

*Figure 4.* **Multi-scale decomposition of feature importance on a volume using WAM**. Coarse scales (yellow) highlight the edges of the number, capturing its global structure. Fine scales (blue) focus on the digit's center, capturing high-frequency details and localized variations.

**Audio: identifying key sound components.** Figure 5 qualitatively illustrates an application of WAM for audio signals. We perform a noise experiment by adding 0 dB white noise to a target audio to form the input audio. The model's prediction remains unchanged after adding noise, indicating it continues to focus on the relevant parts of the input audio. Identifying key parts of the audio input is cru-

*Figure 5.* **Qualitative illustration of WAM for audio in a noise experiment**. We add 0 dB white noise to the audio of the target class ("Crow") as input to the classifier. Audio reconstructed using important wavelet coefficients effectively removes noise and highlights key parts of the target class (marked with a green box). WAM identifies these relevant parts in a fully post-hoc manner.

cial for improving model interpretability and ensuring it focuses on meaningful signal components. Traditionally, this task has been addressed using trained models such as Non-Negative Matrix Factorization (NMF, see for instance Parekh et al., 2022), which requires training to extract relevant features and explain decisions.

In contrast, WAM performs this identification in a post-hoc manner, simplifying the task by directly highlighting the important audio components without the need for prior training. The interpretation audio in Figure 5, generated using top wavelet coefficients, provides further insights into the decision process and is consistent with the fact that the model has not been affected by the addition of noise. Therefore, WAM effectively filters out the corrupted audio

while highlighting the key parts of the audio. Similarly, we discuss in Appendix D.3 how WAM also retrieves the key parts of an audio signal that has been corrupted by another source (overlap experiment).

## 5. Discussion

**Conclusion.** We bring a novel perspective on feature attribution by computing explanations in the wavelet domain rather than the pixel domain, providing a framework applicable to audio, images, and volumes. This method shifts away from traditional pixel-based decompositions used in saliency mapping, offering more precise insights into model decisions by leveraging the wavelet domain's ability to preserve inter-scale dependencies. This ensures that critical aspects like frequency and spatial structures are maintained, resulting in richer explanations compared to traditional feature attribution methods.

Our method, WAM, shows a strong ability to highlight essential audio components in noisy samples, isolate necessary volumes and texture features for accurate predictions, and offer richer explanations for shape classification. Quantitatively, it achieves state-of-the-art results across audio, image, and volume benchmarks.

**Limitations & future works.** Despite its advantages, the current method does not extend to point cloud data. For audio, the greedy extraction of important coefficients is unsuitable for generating listenable explanations.

Future work could explore alternative wavelet decompositions, such as continuous or complex wavelets for audio explanations and graph wavelet transforms to handle unstructured point clouds. Additionally, our method could be applied to videos, mathematically similar to the 3D data used in this work.

The unification of the feature attribution domain finds a natural application in explaining multi-modality models (e.g., Vision-Langage Models). Formally, the wavelet decomposition cannot be applied to text data, thus limiting the applicability of our method in natural language processing applications. Solutions to overcome this difficulty may be found by building on existing methods in this field, which have been surveyed by Lyu et al. (2024).

More broadly, we hope that this work will open the discussion on the properties and expressiveness of the domain in which feature attribution is carried out. In particular, the theoretical properties of expanding feature attribution beyond the pixel space could be further discussed.

## Impact Statement

This work addresses the growing need for grounded, interpretable AI systems, especially in light of increasing regulatory demands for transparency in machine learning. By providing a unified approach to explainability across diverse modalities (audio, image, and volumes) our method ensures that model explanations remain faithful to the inherent structure of the data. In parallel, as multimodal models become more prevalent, our work offers a solution that can be applied to any input type, facilitating more consistent and robust explanations.

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

# A. Implementational details and consistency checks

### A.1. Effect of smoothing and integration on the explanations.

Figure 6 illustrates the improvement of the quality of the explanations by smoothing (third row) or integrating (fourth row) the gradients, compared to their raw values (second row). Smoothing follows Equation (7) and integration follows Equation (8). We can see that both methods display complementary properties regarding the explanation. $\text{WAM}_{SG}$ enables to visualize highlights the important locations within scales, while $\text{WAM}_{IG}$ emphasizes on the relative importance of each scale. This suggests that the variants of WAM can be used for different applications. If one is interested in studying the inter-scale dependencies, the reliance on different frequency ranges, or more broadly, the robustness of the model, then $\text{WAM}_{IG}$ should be preferred. On the other hand, if one wants to focus on the important details at each scale, for instance, to highlight the important patterns for the prediction of a given class, then $\text{WAM}_{SG}$ appears as more relevant.

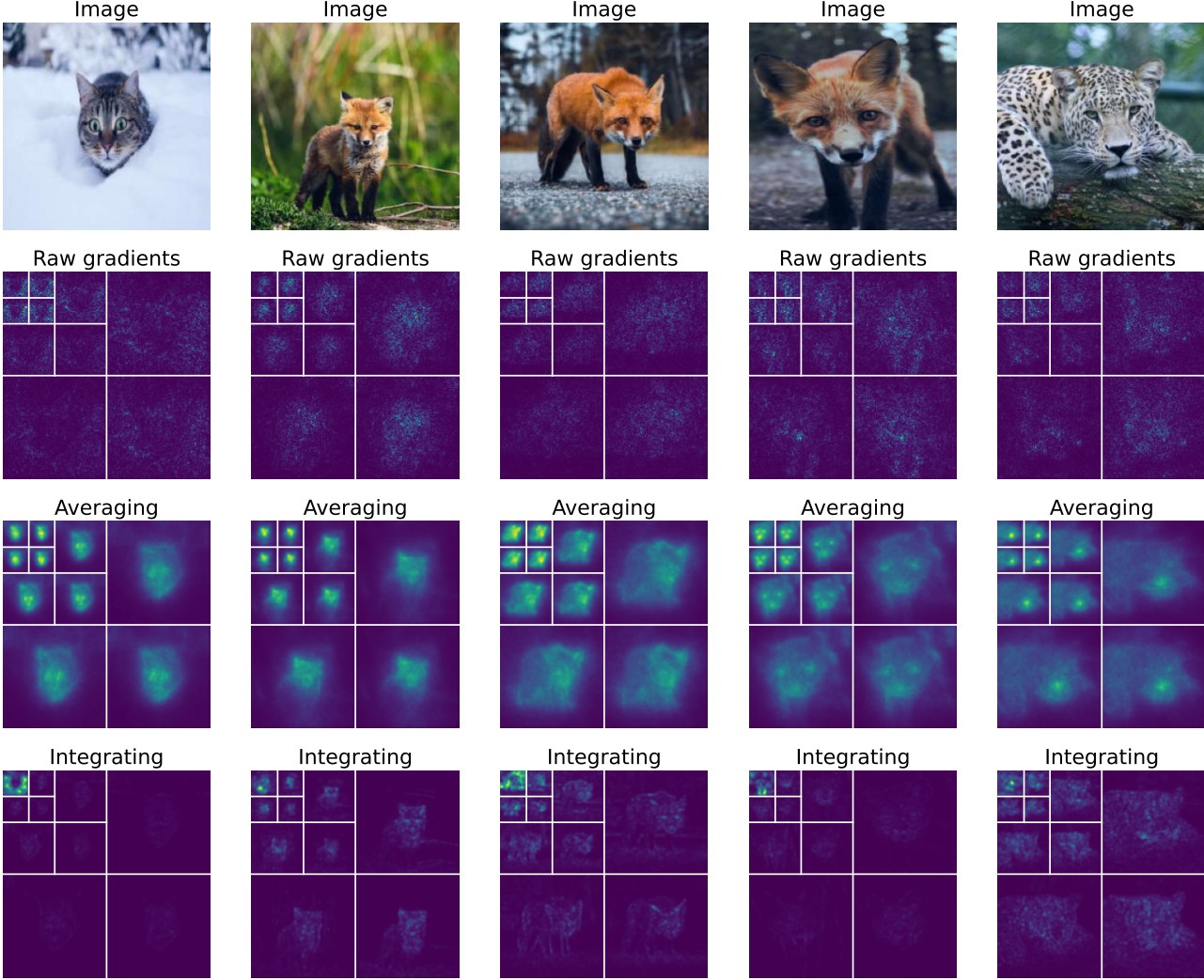

*Figure 6.* **Effect of averaging or smoothing** (3rd row) **and integration** (4th row) compared to the raw gradients (2nd) when computing the WAM of the images (1st row). The explanations are depicted in the wavelet domain. "Smoothing" corresponds to the application of SmoothGrad (Smilkov et al., 2017) and "Integrating" to the application of Integrated Gradients (Sundararajan et al., 2017).

## A.2. Effect of the mother wavelet

**Definition.** The mother wavelet is the fundamental function $\psi \in L^2(\mathbb{R}^n)$ used to generate a family of wavelets through dilation and translation. This construction enables multiresolution analysis. From a mother wavelet $\psi(x)$, the family of function is defined as

$$\psi_{\lambda,b}(x) = \frac{1}{\lambda^{n/2}} \left( \frac{x - b}{\lambda} \right), \tag{10}$$

where $\lambda \in \mathbb{R}^n \setminus \{0\}$ and $b \in \mathbb{R}^n$ are respectively the scale and translation parameters. The choice of the mother wavelet influences which features are captured, the quality of signal reconstruction, and the effectiveness of tasks like compression, denoising, and feature extraction.

**Interpretability.** WAM formally indicates that the model is sensitive to a given region in the space-scale domain. Since the reconstructed image is the same, no matter the chosen basis for decomposition, the interpretation in terms of structural components (i.e., what those coefficients correspond to) depends on the choice of the mother wavelet. The choice of the mother wavelet is therefore grounded in the application of interest, and the information that is invariant to the choice of the mother wavelet is the importance of of a given scale at a given location.

Among the popular mother wavelet, the Haar wavelet (Haar, 1910) enables us to interpret the sensitivity of the model in terms of sharp changes, edges, or discontinuities on the image. It can be used for instance to detect grid-like structures. Daubechies wavelets (Daubechies, 1988) are smoother than Haar wavelets and highlight smooth patterns or fine textural details. Finally, the Biorthogonal (bior) wavelet (Cohen et al., 1992) underlines curves, contours, and symmetric structure, especially on images.

Figure 7 illustrates the different structural features emphasized depending on the choice of the mother wavelet. In this example, we can see that the Haar wavelet decomposes the image in terms of sudden transitions, highlighting the edges of the input image. In contrast, using the Daubechies wavelet reveals more of the image textures. In Appendix D.4 we illustrate the usefulness of WAM in a practical setting, where the chosen mother wavelet was the Haar wavelet, since the object of interest was to understand the model's behavior with respect to the detection of sharp objects on images.

Input Image        Haar Coefficients        Daubechies Coefficients

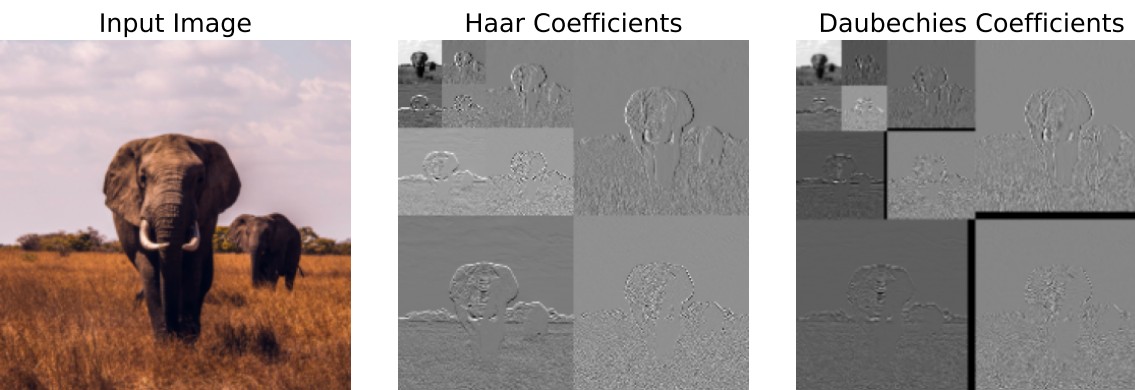

*Figure 7.* **Impact of the mother wavelet.** Different mother wavelets emphasize different features of the image.

**Quantitative checks.** Table 2 empirically checks that the explanations generated by WAM are invariant to the choice of the mother wavelet. For images and volumes, the insertion and deletion scores remain unchanged. Results are reported up to the 3rd decimal, but results remain the same up to the 8th decimal. For audio, the numerical results are close but not identical. This slight variation arises because the reconstruction of the input signal depends on the choice of the mother wavelet. Consequently, different wavelets can lead to minor differences in the results. However, when the reconstructions align, the resulting explanations are identical.

*Table 2.* **Impact of the mother wavelet.** Changing the mother wavelet does not affect the $\text{WAM}_{IG}$ Insertion and Deletion scores for images and volumes, but it causes slight variations for audio.

| | Audio | | Images | | Volumes | |
|---|---|---|---|---|---|---|
| *Model* | *ResNet* | | *ResNet* | | *ResNet* | |
| *Dataset* | *ESC-50* | | *ImageNet* | | *AdrenalMNIST* | |
| | *Insertion* | *Deletion* | *Insertion* | *Deletion* | *Insertion* | *Deletion* |
| Bior | 0.442 | 0.256 | 0.422 | 0.078 | 0.404 | 0.631 |
| Daubechies | 0.446 | 0.266 | 0.422 | 0.078 | 0.404 | 0.631 |
| Haar (baseline) | 0.438 | 0.263 | 0.422 | 0.078 | 0.404 | 0.631 |

### A.3. Randomization checks

**Motivation.** The sanity checks introduced by Adebayo et al. (2018) aim at assessing whether an explanation depends on the model's parameters and the input labels. These tests assess the faithfulness of an explanation beyond visual evaluation. The randomization test evaluates whether an explanation depends on the model's parameters. Parameters have a strong effect on a model's performance. Therefore, for a saliency method to be useful for debugging or analyzing a model, it should be sensitive to its parameters. Adebayo et al. (2018) proposed different methods to randomize the model parameters. One particularly interesting implementation is the "cascading" randomization, in which the weights are randomized from the top to the bottom layers.

**Method.** We compute WAM for 1,000 ImageNet validation samples for a set of increasingly randomized models. A randomized layer is a layer that we reset at its initial value. We consider a ResNet-18 and randomize its layers from the shallowest `conv1` to the deepest `fc`. We then compute the rank correlation (or Pearson correlation coefficient, Pearson, 1896) between the WAM of the original, fully-trained model (labeled `orig`) and the randomized models (labeled by the name of the layer until which they are randomized).

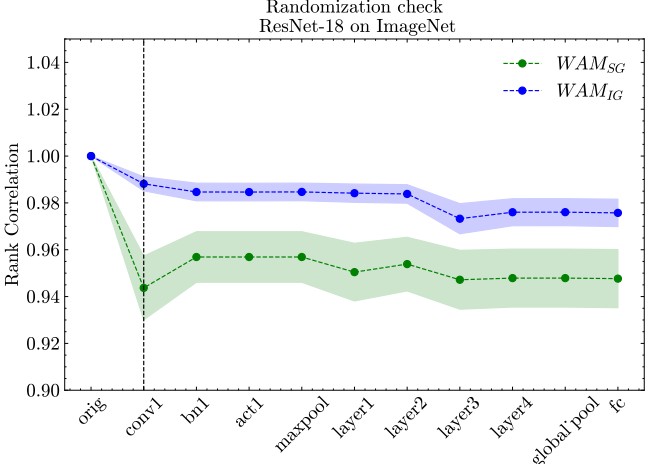

*Figure 8.* **Cascading randomization** of WAM for explaining a ResNet-18 on ImageNet. The $y$ axis indicates the rank correlation between the original explanation and the explanation derived for randomization up that layer. The rank correlation is averaged over 1,000 ImageNet validation images.

**Results.** Figure 8 presents the results. The dotted line represents the average rank correlation across the 1,000 images, and the intervals represent the 95% confidence intervals. We can see that the correlation between WAM s significantly decreases as the randomization increases, thereby showing that WAM is sensitive to the model's parameters. The lower decrease that we observe for $\text{WAM}_{IG}$ compared to $\text{WAM}_{SG}$ comes from the fact that $\text{WAM}_{IG}$ reflects more the inter-scales distribution of the importance than $\text{WAM}_{SG}$ does. As pointed out by Rahaman et al. (2019) and Yin et al. (2019), even random models

exhibit a spectral bias, i.e., a natural tendency to favor lower frequencies over higher ones, which translates here into the fact of naturally putting more importance on coarser scales rather than finer scales, no matter the depth of the randomization. Figure 9 qualitatively illustrates how the explanations evolve as the randomization depth increases.

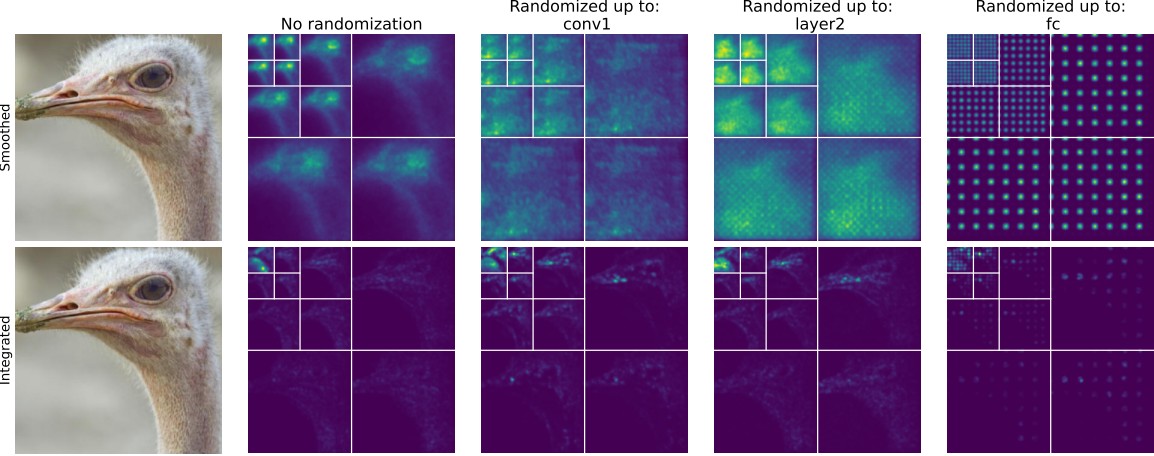

*Figure 9.* **Illustration of the randomization test as a cascade of randomizations of the layers of the classifier.** From left to right shows the explanation from WAM for an increasingly randomized ResNet-18.

## B. Theoretical properties

Sundararajan et al. (2017) highlighted two axioms that feature attribution methods should satisfy, namely sensitivity and implementation invariance. We first recall the definitions of these axioms.

**Axiom B.1** (Sensitivity (a)). *If two inputs $\boldsymbol{x}$ and $\boldsymbol{x}'$ differ in one feature and have different predictions, then the differing feature must receive a non-zero attribution.*

**Axiom B.2** (Implementation Invariance). *Two functionally equivalent networks (i.e., computing the same function) must yield the same attributions.*

Since $\text{WAM}_{IG}$ builds on integrated gradients, we may wonder whether it inherits the properties of the original integrated gradients. In the following, let $\boldsymbol{f}_c : \mathbb{R}^d \to \mathbb{R}$ be the class logit function in input space $\mathcal{W}$ denotes the wavelet transform operator and $\mathcal{W}^{-1}$ its inverse. We define $\boldsymbol{z} = \mathcal{W}(\boldsymbol{x})$ and let $\boldsymbol{z}_0$ be a fixed baseline in the wavelet space. We also define the composed function $f = \boldsymbol{f}_c \circ \mathcal{W}^{-1}$, such that $f(\boldsymbol{z}) = \boldsymbol{f}_c(\boldsymbol{x})$.

### B.1. $\text{WAM}_{IG}$ satisfies completeness

We can show that under some conditions on the choice of the mother wavelet $\psi$, $\text{WAM}_{IG}$ satisfies completeness, and thus sensitivity. Following Sundararajan et al. (2017), this implies that $\text{WAM}_{IG}$ satisfies sensitivity (a).

**Definition B.1** (Completeness). *An attribution method $\boldsymbol{\gamma}$ satisfies* completeness *if*

$$\sum_{i=1}^{n} \boldsymbol{\gamma}(\boldsymbol{x})_i = F(\boldsymbol{x}) - F(\boldsymbol{x}_0). \tag{11}$$

*for any given $F : \mathbb{R}^n \to \mathbb{R}$ differentiable almost everywhere.*

**Theorem B.1.** *If $\mathcal{W}^{-1}$ is differentiable almost everywhere, then $\text{WAM}_{IG}$ satisfies completeness:*

$$\sum_{i=1}^{n} \text{WAM}_{IGi}(\boldsymbol{z}) = \boldsymbol{f}_c(\boldsymbol{x}) - \boldsymbol{f}_c(\boldsymbol{x}_0) \tag{12}$$

*Proof.* Proposition 1 in Sundararajan et al. (2017) states that for some $f$ differentiable almost everywhere,

$$\sum_{i=1}^{n} IG_i(\boldsymbol{x}) = f(\boldsymbol{x}) - f(\boldsymbol{x}_0)$$

since by definition, $\text{WAM}_{IG}$ is IG applied to $f(\boldsymbol{z})$, we have

$$\sum_{i=1}^{n} \text{WAM}_{IGi}(\boldsymbol{z}) = f(\boldsymbol{z}) - f(\boldsymbol{z}_0). \qquad (\star)$$

Now taking $f = \boldsymbol{f}_c \circ \mathcal{W}^{-1}$, Equation $(\star)$ can be re-written as

$$\begin{aligned}
\sum_{i=1}^{n} \text{WAM}_{IGi}(\boldsymbol{z}) &= f(\boldsymbol{z}) - f(\boldsymbol{z}_0) \\
&= \boldsymbol{f}_c(\mathcal{W}^{-1}(\boldsymbol{z})) - \boldsymbol{f}_c(\mathcal{W}^{-1}(\boldsymbol{z}_0)) \\
&= \boldsymbol{f}_c(\boldsymbol{x}) - \boldsymbol{f}_c(\boldsymbol{x}_0)
\end{aligned}$$

and $\boldsymbol{f}_c \circ \mathcal{W}^{-1}$ remains differentiable (almost everywhere) provided $\mathcal{W}^{-1}$ is differentiable. This can be enforced by choosing a mother wavelet $\psi$ that is smooth (e.g., with Daubechies but not with Haar wavelets). Thus, provided $\psi$ is smooth, $\text{WAM}_{IG}$ satisfies completeness. $\qquad\square$

### B.2. Additional theoretical properties

It is also possible to show that $\text{WAM}_{IG}$ satisfies implementation invariance and linearity. As with completeness, it is enough to show that the corresponding composed function $f$ "respects" implementation invariance or linearity of $\boldsymbol{f}_c$. And since $IG$ satisfies both these properties, $\text{WAM}_{IG}$ will satisfy as well as it is IG applied to the corresponding $f$ functions.

For implementation invariance, if $\boldsymbol{f}_c^{(1)}, \boldsymbol{f}_c^{(2)}$ have different architectures but exactly same output for any input $x$, the corresponding composed functions $f_{(1)}, f_{(2)}$ also have the same output for any input $z$ and thus $\text{WAM}_{IG}$ outputs are exactly the same for any given input $x$ for $\boldsymbol{f}_c^{(1)}, \boldsymbol{f}_c^{(2)}$.

For linearity, one needs to show that $\text{WAM}_{IG}$ output for a linear combination of any two functions $\boldsymbol{f}_c^{(3)} = a\boldsymbol{f}_c^{(1)} + b\boldsymbol{f}_c^{(2)}$ is linear combination of $\text{WAM}_{IG}$ output for each of them. Let $f_{(i)}$ be the associated composed function of $\boldsymbol{f}_c^{(i)}$. Then $f_{(3)} = af_{(1)} + bf_{(2)}$ and because of linearity of $IG$ applied on $f_{(i)}$'s, $\text{WAM}_{IG}$ satisfies linearity as well. Thus, it's easy to see that $\text{WAM}_{IG}$ also satisfies implementation invariance and linearity axioms. More generally, we believe that $\text{WAM}_{IG}$ is a path method, and thus inherits the properties of the path methods proved by Friedman (2004). However, the focus of the paper is on the proposal and practical usage of wavelet attribution methods. We keep any formal study of theoretical properties of $\text{WAM}_{IG}$ or other variants of WAM as future work.

### B.3. Feature attribution beyond the input space

More generally, our approach expands attribution methods on transformations of the input in an alternative domain. It seems that the properties satisfied by traditional attribution method generalize to the case where attribution is done on transformations of the input data. This work thus opens a promising direction in the systematic study of generalized attribution spaces, or transformed domains where gradients can be computed more expressively while preserving fidelity to the original input.

## C. Complements on the quantitative evaluation

### C.1. Benchmark construction

**Images.** Our models' parameterizations for benchmarking WAM on images are the following:

- ResNet (He et al., 2016): unless specified otherwise, we consider the `resnet18` variant,

- EfficientNet (Tan & Le, 2019): we consider the `tf_efficientnet_b0.ns_jft_in1k` variant,

- ConvNext (Liu et al., 2022): we consider the `convnext_small.fb_in22k_ft_in1k_384` variant,

- DeiT (Touvron et al., 2021): we consider the `deit_tiny_patch16_224.fb_in1k`

All models are retrieved from the PyTorch Image Models (Wightman, 2019) repository. We load the model with the pretrained weights and directly evaluate them on the validation set of ImageNet. We implement the SmoothGrad, GradCAM, and GradCAM Plus Plus methods ourselves and use the Captum library (Kokhlikyan et al., 2020) for implementing the Intergrated Gradients and the Saliency methods.

**Audio.** For audio, we use the same technical infrastructure to evaluate our method. We use the CNN classification model of Kumar et al. (2018) as our black-box model to explain. We consider a single model as alternative models (Huang & Leanos, 2018; Wilkinghoff, 2021; Lopez-Meyer et al., 2021) are only variations around the same topology. We add 0 dB white noise to the ESC-50 samples using the pseudocode displayed in Figure 10.

```
1  # Input: audio (input audio signal, array of int16)
2  # Output: noisy_audio (audio with added Gaussian noise, array of int16)
3
4  def add_gaussian_noise(audio):
5    # Convert the audio to float32 for safe computation
6    audio_float = convert_to_float32(audio)
7
8    # Calculate RMS (Root Mean Square) of the audio signal
9    rms_signal = sqrt(mean(audio_float ** 2))
10
11   # Generate Gaussian noise
12   noise = random_normal_distribution(mean=0,
13                                      std=1,
14                                      shape=audio_float.shape)
15
16   # Calculate RMS of the generated noise
17   rms_noise = sqrt(mean(noise ** 2))
18
19   # Scale noise to have the same RMS as the audio signal
20   noise = noise * (rms_signal / rms_noise)
21
22   # Add noise to the audio signal
23   noisy_audio_float = audio_float + noise
24
25   # Clip the noisy audio to ensure it stays within the int16 range
26   noisy_audio_clipped = clip(noisy_audio_float, -32768, 32767)
27
28   # Convert the clipped noisy audio back to int16
29   noisy_audio = convert_to_int16(noisy_audio_clipped)
30
31   return noisy_audio
```

*Figure 10.* **Pseudo-code for adding Gaussian noise to audio.**

**Volumes.** For volumes, we consider the LATEC benchmark (Klein et al., 2024) and evaluate our method on two datasets of MedMNIST (Yang et al., 2023): AdrenalMNIST3D and VesselMNIST. We carry out our evaluation over the complete test set. We included WAM as an additional feature attribution method for fair comparisons with existing methods.

### C.2. Definition of the evaluation metrics

**Insertion and Deletion.** Insertion and deletion are two evaluation metrics proposed by Petsiuk et al. (2018). These metrics are "area-under-curve" (AUC) metrics, which report the change in the predicted probability for the image class when inserting (resp. removing) meaningful information highlighted by the attribution method. Petsiuk et al. (2018) initially defined this metric for images, where the important features correspond to pixels. We expand this metric to wavelet coefficients, thus enabling computation of the Insertion and the Deletion for any modality.

Both metrics consider an input in a baseline state. Insertion consists in adding the most important features identified by the attribution method. Formally, at step $k$ with a subset $u_k$ of important features (which correspond in our case in wavelet coefficients) at step $k$,

$$\text{Insertion}^{(k)} = \boldsymbol{f}(\boldsymbol{x}_{[\boldsymbol{x}_{-u_k}=\boldsymbol{x}_0]}), \tag{13}$$

where $\boldsymbol{f}(.)$ is the predicted probability and $-u$ denotes the complementary set of $u$. We add features by decreasing order of importance and for $k_1 \leq k_2, u_{k_1} \subseteq u_{k_2}$, i.e., we gradually add more and more features until we eventually recover the full input $\boldsymbol{x}$.

The deletion performs the opposite operation where we start from a plain input with all variables and we gradually set features in the baseline state $\boldsymbol{x}_0$, from the most important to the less important. We have

$$\text{Deletion}^{(k)} = \boldsymbol{f}(\boldsymbol{x}_{[\boldsymbol{x}_{u_k}=\boldsymbol{x}_0]}). \tag{14}$$

Finally, for the insertion and the deletion, we measure the AUC, which is comprised between 0 and 1. Given $K$ steps, the Insertion score of the feature attribution $\boldsymbol{\gamma}$ for the model $\boldsymbol{f}$ is

$$\text{Ins}(\boldsymbol{f}, \boldsymbol{\gamma}) = \sum_{k=1}^{K} \text{Insertion}^{(k)} \Delta_k = \sum_{k=1}^{K} \boldsymbol{f}(\boldsymbol{x}_{[\boldsymbol{x}_{-u_k}=\boldsymbol{x}_0]}) \Delta_k, \tag{15}$$

where $\Delta_k$ is the width between two subintervals. The computation is analogous for $\text{Del}(\boldsymbol{f}, \boldsymbol{\gamma})$.

If the attribution method picks relevant features, then only including them (resp. only removing them) should result in a large increase (resp. large decrease) in the predicted probability. Therefore, the AUC should be close to 1 for the insertion and close to 0 for the deletion. We set the baseline to $\boldsymbol{x}_0 = 0$.

$\mu$**-Fidelity.** The $\mu$-Fidelity is a correlation metric. It measures the correlation between the decrease of the predicted probabilities when features are in a baseline state and the importance of these features. We have

$$\mu\text{-Fidelity} = \underset{\substack{u \subseteq \{1,\ldots,K\}, \\ |u|=d}}{\text{Corr}} \left( \sum_{i \in u} \boldsymbol{g}(\boldsymbol{x}_i), \boldsymbol{f}(\boldsymbol{x}) - \boldsymbol{f}(\boldsymbol{x}_{\boldsymbol{x}_u=\boldsymbol{x}_0}) \right), \tag{16}$$

where $\boldsymbol{g}$ is the explanation function (i.e., the explanation method), which quantifies the importance of the set of features $u$.

**Faithfulness on Spectra.** The Faithfulness on Spectra (**FF**, Parekh et al., 2022) measures how important is the generated interpretation for a classifier. The metric is calculated by measuring the drop in class-specific logit value $\boldsymbol{f}(\boldsymbol{x})_c$ when the masked out portion of the interpretation mask $\boldsymbol{m}_{\boldsymbol{\gamma}}$ is input to the classifier. This amounts to calculating,

$$\text{FF}_{\boldsymbol{x}} = \boldsymbol{f}(\boldsymbol{x})_c - \boldsymbol{f}\left(\boldsymbol{x} \odot (\boldsymbol{1} - \boldsymbol{m}_{\boldsymbol{\gamma}})\right)_c. \tag{17}$$

It should be noted that this strategy to simulate removal may introduce artifacts in the input that can affect the classifier's output unpredictably. Also, interpretations on samples with poor fidelity can lead to negative $\text{FF}_{\boldsymbol{x}}$. These observations point to this metric's potential instability and outlying values. Thus, we report the final faithfulness of the system as the median of $\text{FF}_{\boldsymbol{x}}$ over the test set, denoted by FF. A positive FF would signify that interpretations are faithful to the classifier.

**Input Fidelity.** The Input Fidelity (**Fid-In**, Paissan et al., 2023) measures if the classifier outputs the same class prediction on the masked-in portion of the input image. It is defined as,

$$\text{Fid-In} = \frac{1}{n} \sum_{i=1}^{n} \mathbb{I} \left[ \arg\max_c \boldsymbol{f}(\boldsymbol{x}_i)_c = \arg\max_c \boldsymbol{f}_c(\boldsymbol{x}_i \odot \boldsymbol{m}_{\boldsymbol{\gamma}}) \right], \tag{18}$$

where $\mathbb{I}$ denotes the indicator function and, again, larger values are better.

## C.3. Additional evaluation results

This section provides additional quantitative results. For audio, we evaluated WAM using additional metrics and on a noisy version of the dataset ESC-50 and SONY-UST, as done by Parekh et al. (2022). We also evaluated WAM by computing explanations from the mel-spectrogram and the wavelet coefficients of the input audio. For images, we evaluated our method using different model topologies: ResNet, EfficientNet, Data Efficient Transformer, and ConvNext. We also included additional metrics and additional methods. For volumes, we evaluated the performance of our method on MNIST3D using a ResNet3D backbone and we included two additional datasets, AdrenalMNIST and VesselMNIST, and three model topologies: ResNet3D, EfficientNet3D, and 3D Former. We also explored the effect of the number of decomposition levels on performance. Across all these variants, WAM demonstrated solid performance compared to the baselines, showing its robustness and versatility across use cases and datasets.

### AUDIO

Table 3 presents the full evaluation results for all audio metrics. We report the results for the clean and noisy dataset, as done by Parekh et al. (2022). For WAM, we consider two variants: when the explanations are computed on the mel-spectrogram of the input audio and when it is computed from the wavelet coefficients. We can see that when the explanations is computed from the wavelet coefficients, accuracy is higher. We report the results on two datasets, ESC-50 (Piczak, 2015) and SONY-USC (Cartwright et al., 2019)

*Table 3.* **Evaluation scores** of WAM and comparison with baselines on 400 audio samples from ESC-50 (fold 1) and on the validation set of the SONY-UST dataset. The column "-" indicates that the samples are unaltered. The column "+WN" indicates that the samples have 0 dB Gaussian white noise. **Bolded** results are the best and underlined values are the second best.

| | Metric | Faithfulness (↑) | | Insertion (↑) | | Deletion(↓) | | FF (↑) | | Fid-In (↑) | |
|---|---|---|---|---|---|---|---|---|---|---|---|
| | | - | +WN | - | +WN | - | +WN | - | +WN | - | +WN |
| *ESC-50* | IntegratedGradients | **0.264** | **0.310** | 0.267 | 0.312 | **0.047** | **0.045** | **0.207** | **0.207** | 0.220 | 0.225 |
| | GradCAM | 0.072 | 0.073 | 0.274 | 0.274 | 0.201 | 0.201 | 0.137 | 0.135 | 0.517 | 0.542 |
| | Saliency | 0.066 | 0.065 | 0.220 | 0.221 | 0.154 | 0.156 | 0.166 | 0.168 | 0.253 | 0.245 |
| | SmoothGrad | 0.184 | 0.184 | 0.251 | 0.251 | 0.067 | 0.067 | 0.193 | 0.194 | 0.177 | 0.175 |
| | WAM$_{SG}$ (ours, wavelet) | 0.197 | 0.205 | **0.449** | **0.452** | 0.252 | 0.246 | 0.132 | 0.130 | **0.718** | **0.690** |
| | WAM$_{IG}$ (ours, wavelet) | 0.176 | 0.182 | 0.436 | 0.442 | 0.260 | 0.261 | 0.118 | 0.124 | 0.652 | 0.657 |
| | WAM$_{SG}$ (ours, melspec) | 0.009 | 0.007 | 0.169 | 0.166 | 0.159 | 0.161 | 0.152 | 0.149 | 0.117 | 0.122 |
| | WAM$_{IG}$ (ours, melspec) | 0.000 | 0.004 | 0.168 | 0.171 | 0.168 | 0.167 | 0.149 | 0.149 | 0.105 | 0.128 |
| *SONY-UST* | IntegratedGradients | **0.151** | **0.151** | **0.732** | **0.732** | 0.581 | 0.582 | -0.122 | -0.121 | 0.000 | 0.002 |
| | GradCAM | 0.027 | 0.026 | 0.600 | 0.598 | 0.573 | 0.571 | -0.004 | -0.004 | 0.800 | 0.774 |
| | Saliency | 0.000 | 0.003 | 0.706 | 0.708 | 0.706 | 0.705 | -0.224 | -0.225 | 0.000 | 0.002 |
| | SmoothGrad | 0.115 | 0.116 | 0.682 | 0.682 | **0.567** | **0.566** | -0.098 | -0.098 | 0.000 | 0.002 |
| | WAM$_{SG}$ (ours, wavelet) | -0.076 | -0.076 | 0.598 | 0.600 | 0.674 | 0.675 | 0.029 | 0.026 | **0.882** | **0.878** |
| | WAM$_{IG}$ (ours, wavelet) | -0.075 | -0.080 | 0.591 | 0.591 | 0.666 | 0.671 | **0.037** | **0.036** | 0.866 | 0.840 |
| | WAM$_{SG}$ (ours, melspec) | 0.003 | 0.010 | 0.677 | 0.679 | 0.674 | 0.669 | -0.188 | -0.183 | 0.000 | 0.002 |
| | WAM$_{IG}$ (ours, melspec) | 0.028 | 0.023 | 0.728 | 0.723 | 0.700 | 0.700 | -0.202 | -0.200 | 0.000 | 0.000 |

### IMAGES

For images, we report the evaluation results across different model topologies and with additional methods. On a ResNet, we also compare our method with more recent feature attribution methods. All explanations are computed on 1,000 samples from the validation set of ImageNet.

**Faithfulness and $\mu$-Fidelity.** Table 4 reports the evaluation results using different metrics across a wide range of topologies. We also include additional methods such as GradCAM ++ (Selvaraju et al., 2017) and Guided Brackpropagation

(Springenberg et al., 2014). We can see that in terms of Faithfulness, our method outperforms alternative approaches over all model topologies. On the other hand, the performance measured by the $\mu$-Fidelity aligns WAM with existing approaches. We can see that the good results reported in Table 4 are mostly driven by our method's very good results in Insertion. We can see that WAM systematically outperforms the other methods in both Insertion and Deletion.

*Table 4.* **Evaluation results of WAM for images.** We report the **Faithfulness** (Muzellec et al., 2024), $\mu$**-Fidelity** (Bhatt et al., 2021) and **Insertion** and **Deletion** (Petsiuk et al., 2018) scores obtained on 1,000 images from the validation set of ImageNet and for different model architectures. **Bolded** results are the best and underlined values are the second best.

| | Model | ResNet | ConvNext | EfficientNet | DeiT | Mean |
|---|---|---|---|---|---|---|
| | Saliency | 0.025 | 0.032 | 0.008 | 0.038 | 0.025 |
| | Integrated Gradients | 0.000 | 0.001 | 0.000 | 0.003 | 0.001 |
| | GradCAM | 0.134 | 0.072 | 0.061 | 0.162 | 0.107 |
| | GradCAM++ | 0.184 | 0.055 | 0.050 | 0.044 | 0.083 |
| *Faithfulness (↑)* | SmoothGrad | 0.023 | 0.000 | 0.010 | 0.004 | 0.009 |
| | Guided-Backpropagation | 0.001 | 0.001 | 0.001 | 0.000 | 0.000 |
| | WAM$_{SG}$ (ours) | **0.438** | **0.334** | **0.350** | **0.423** | **0.386** |
| | WAM$_{IG}$ (ours) | 0.344 | 0.359 | 0.370 | 0.420 | 0.373 |
| | Saliency | 0.154 | 0.186 | 0.180 | 0.195 | 0.179 |
| | Integrated Gradients | **0.228** | 0.223 | **0.219** | **0.241** | **0.228** |
| | GradCAM | 0.141 | 0.216 | 0.149 | 0.151 | 0.164 |
| | GradCAM ++ | 0.135 | 0.212 | 0.141 | 0.222 | 0.178 |
| *$\mu$-Fidelity (↑)* | SmoothGrad | 0.220 | 0.227 | 0.211 | 0.230 | 0.222 |
| | Guided Backpropagation | 0.216 | 0.229 | **0.234** | 0.226 | 0.226 |
| | WAM$_{SG}$ (ours) | 0.215 | 0.208 | 0.213 | 0.216 | 0.213 |
| | WAM$_{IG}$ (ours) | 0.170 | 0.166 | 0.165 | 0.182 | 0.171 |
| | Saliency | 0.134 | 0.381 | 0.148 | 0.194 | 0.214 |
| | Integrated Gradients | 0.087 | 0.305 | 0.113 | 0.095 | 0.150 |
| | GradCAM | 0.413 | 0.495 | 0.364 | 0.352 | 0.406 |
| | GradCAM ++ | 0.452 | 0.562 | 0.350 | 0.313 | 0.419 |
| *Insertion (↑)* | SmoothGrad | 0.106 | 0.253 | 0.129 | 0.108 | 0.149 |
| | Guided Backpropagation | 0.090 | 0.332 | 0.117 | 0.093 | 0.158 |
| | WAM$_{SG}$ (ours) | **0.557** | **0.606** | **0.447** | **0.546** | **0.539** |
| | WAM$_{IG}$ (ours) | 0.422 | 0.557 | 0.419 | 0.492 | 0.473 |
| | Saliency | 0.109 | 0.349 | 0.140 | 0.156 | 0.189 |
| | Integrated Gradients | 0.087 | 0.304 | 0.113 | 0.092 | 0.149 |
| | GradCAM | 0.279 | 0.423 | 0.303 | 0.190 | 0.299 |
| | GradCAM ++ | 0.268 | 0.507 | 0.300 | 0.269 | 0.336 |
| *Deletion (↓)* | SmoothGrad | 0.083 | 0.253 | 0.119 | 0.104 | 0.140 |
| | Guided Backpropagation | 0.089 | 0.331 | 0.116 | 0.092 | 0.157 |
| | WAM$_{SG}$ (ours) | 0.119 | 0.272 | 0.097 | 0.123 | 0.153 |
| | WAM$_{IG}$ (ours) | **0.078** | **0.198** | **0.049** | **0.072** | **0.099** |

**Additional methods.** Table 5 compares WAM with more recent feature attribution methods, considering a ResNet as a model backbone. We compare WAM against LayerCAM (Jiang et al., 2021), Guided Backpropagation (Springenberg et al., 2014), LRP (Binder et al., 2016) and SRD (Han et al., 2024).

*Table 5.* **Evaluation of WAM on ImageNet** with recent feature attribution methods using a ResNet backbone. Best results are **bolded**, second best underlined.

| Metric | Faithfulness (↑) | Insertion (↑) | Deletion (↓) | $\mu$-Fidelity (↑) |
|---|---|---|---|---|
| Guided Backprop | 0.001 | 0.090 | 0.089 | **0.216** |
| LayerCAM | 0.139 | 0.432 | 0.293 | 0.143 |
| LRP | 0.039 | 0.277 | 0.238 | 0.126 |
| SRD | 0.002 | 0.102 | 0.101 | 1.151 |
| $\text{WAM}_{IG}$ (ours) | **0.438** | **0.557** | 0.119 | 0.215 |
| $\text{WAM}_{SG}$ (ours) | 0.344 | 0.344 | **0.078** | 0.170 |

VOLUMES

**Complementary results on MNIST3D.** In Table 6 we evaluate our method against Saliency (Zeiler & Fergus, 2014a), GradCAM (Selvaraju et al., 2017), GradCAM++ (Chattopadhay et al., 2018), IntegratedGradients (Sundararajan et al., 2017), GuidedBackpropagation (Springenberg et al., 2014) and SmoothGrad (Smilkov et al., 2017) on the dataset MNIST3D. We report the Insertion, the Deletion, and the Faithfulness. Overall, WAM performs as well as or better than the other methods.

*Table 6.* **Evaluation scores (3D) on the full test set of MNIST3D.** Best results are **bolded**, second best underlined.

| Metric | Faithfulness (↑) | Insertion (↑) | Deletion(↓) |
|---|---|---|---|
| Saliency | -0.166 | **0.108** | 0.274 |
| GradCAM | -0.225 | 0.106 | 0.330 |
| GradCAM++ | -0.250 | 0.104 | 0.354 |
| IntegratedGradients | -0.123 | **0.108** | 0.232 |
| GuidedBackprop | -0.110 | 0.108 | 0.218 |
| SmoothGrad | -0.190 | 0.107 | 0.297 |
| $\text{WAM}_{SG}$ (ours, $J = 2$) | -0.153 | **0.108** | 0.261 |
| $\text{WAM}_{IG}$ (ours, $J = 2$) | **-0.094** | 0.106 | **0.200** |
| $\text{WAM}_{SG}$ (ours, $J = 1$) | -0.152 | **0.108** | 0.260 |
| $\text{WAM}_{IG}$ (ours, $J = 1$) | -0.096 | 0.107 | 0.203 |

**Evaluation on MedMNIST.** MedMNIST (Yang et al., 2023) is a collection of standardized biomedical 2D and 3D images. We evaluated WAM on two datasets from MedMNIST, AdrenalMNIST and VesselMNIST. Table 7 shows the evaluation results of WAM on AdrenalMNIST and VesselMNIST using different model topologies.

*Table 7.* **Evaluation scores (3D) on the full test set of AdrenalMNIST and VesselMNIST**. Best results are **bolded**, second best underlined.

| | Model | ResNet3D | | | EfficientNet3D | | | 3D Former | | |
|---|---|---|---|---|---|---|---|---|---|---|
| | Metric | Faith (↑) | Ins (↑) | Del (↓) | Faith (↑) | Ins (↑) | Deletion (↓) | Faith (↑) | Ins (↑) | Del (↓) |
| *AdrenalMNIST* | Saliency | -0.136 | 0.633 | 0.769 | -0.085 | 0.679 | 0.764 | -0.027 | 0.715 | 0.742 |
| | GradCAM | -0.195 | 0.529 | 0.724 | -0.052 | 0.693 | 0.745 | -0.065 | 0.679 | 0.744 |
| | GradCAM++ | -0.137 | 0.531 | 0.668 | -0.076 | 0.695 | 0.771 | -0.091 | 0.656 | 0.748 |
| | IntegratedGradients | -0.204 | 0.531 | 0.735 | -0.056 | 0.705 | 0.761 | -0.077 | 0.666 | 0.743 |
| | Guided Backpropagation | -0.132 | **0.585** | 0.717 | 0.003 | 0.683 | 0.680 | -0.004 | 0.685 | 0.689 |
| | SmoothGrad | **0.115** | **0.663** | 0.548 | -0.070 | 0.686 | 0.756 | -0.051 | 0.680 | 0.731 |
| | WAM$_{SG}$ (ours, $J=2$) | -0.279 | 0.467 | 0.746 | 0.049 | 0.666 | 0.617 | 0.070 | 0.718 | 0.648 |
| | WAM$_{IG}$ (ours, $J=2$) | -0.224 | 0.407 | **0.631** | **0.086** | 0.684 | **0.598** | **0.106** | **0.719** | **0.613** |
| | WAM$_{SG}$ (ours, $J=1$) | -0.273 | 0.474 | 0.746 | 0.026 | 0.661 | 0.635 | 0.050 | 0.717 | 0.667 |
| | WAM$_{IG}$ (ours, $J=1$) | -0.224 | 0.478 | 0.702 | 0.013 | **0.699** | 0.686 | 0.026 | 0.697 | 0.671 |
| *VesselMNIST* | Saliency | -0.068 | 0.833 | 0.901 | -0.747 | **0.131** | 0.878 | -0.099 | 0.762 | 0.861 |
| | GradCAM | -0.095 | 0.789 | 0.884 | -0.728 | 0.126 | 0.854 | -0.065 | 0.804 | 0.869 |
| | GradCAM++ | 0.011 | 0.871 | 0.860 | -0.725 | 0.124 | 0.849 | -0.066 | 0.797 | 0.863 |
| | IntegratedGradients | -0.116 | 0.771 | 0.887 | -0.743 | 0.127 | 0.870 | -0.060 | 0.808 | 0.868 |
| | GuidedBackprop | -0.229 | 0.654 | 0.883 | -0.717 | 0.121 | 0.838 | 0.002 | 0.843 | 0.841 |
| | SmoothGrad | **0.137** | **0.861** | **0.724** | -0.674 | 0.130 | 0.804 | **0.045** | **0.851** | 0.806 |
| | WAM$_{SG}$ (ours, $J=2$) | -0.054 | 0.848 | 0.902 | -0.674 | 0.126 | 0.800 | 0.020 | 0.798 | **0.778** |
| | WAM$_{IG}$ (ours, $J=2$) | -0.064 | 0.756 | 0.820 | **-0.630** | 0.126 | **0.756** | 0.010 | 0.793 | 0.783 |
| | WAM$_{SG}$ (ours, $J=1$) | -0.070 | 0.832 | 0.902 | -0.692 | 0.128 | 0.820 | 0.001 | 0.799 | 0.798 |
| | WAM$_{IG}$ (ours, $J=1$) | -0.069 | 0.787 | 0.856 | -0.691 | 0.128 | 0.819 | -0.070 | 0.764 | 0.834 |

# D. Additional use cases

## D.1. Revisiting meaningful perturbations

**Meaningful perturbations in the wavelet domain.** Meaningful perturbations (Fong & Vedaldi, 2017; Fong et al., 2019) identify the most relevant regions of an input for a model's prediction by learning an optimal mask that hides as little of the image as possible while significantly altering the model's output. This is done by optimizing a mask through gradient-based updates to minimize the classifier's confidence while maintaining a smoothness constraint. The result highlights the most important features in a way that is more structured and localized than saliency maps. We revisit this framework by proposing to recast the problem in the wavelet domain as a more suitable space for optimization. The wavelet domain inherently captures spatial and spectral information, providing a natural structure for producing meaningful and interpretable solutions. Specifically, we solve the following optimization problem:

$$m^\star = \underset{m \in [0,1]^{|\mathcal{X}|}}{\arg\min} \; f_c\left(\mathcal{W}^{-1}(z \odot m)\right) + \alpha \|m\|_1, \tag{19}$$

where $f_c$ represents the classification score, $\mathcal{W}^{-1}$ is the inverse wavelet transform, $z$ is the wavelet transform of the input signal, $\odot$ denotes element-wise multiplication, and $\alpha$ controls the sparsity of the mask $m$. To solve this problem, we initialize the mask as $m_0 = 1$ (i.e., a mask that retains all coefficients) and iteratively update it using gradient descent:

$$m_{i+1} = m_i - \eta \nabla_{m_i}\left(f_c\left(\mathcal{W}^{-1}(z \odot m_i)\right) + \alpha\|m_i\|_1\right), \tag{20}$$

where $\eta$ represents the step size and the gradient is taken with respect to $m_i$. The optimization process continues until convergence is achieved. We call the resulting image the *minimal* image. In practice, we employ the Nadam (Dozat, 2016) optimizer, which combines the benefits of Nesterov acceleration and Adam optimization. Our approach consistently

produces masks with controllable sparsity levels up to 90%, meaning that 90% of the wavelet coefficients are zeroed out, *while* maintaining a classification score comparable to or better than the original prediction. This high sparsity level suggests that the model's decision may rely on a minimal subset of wavelet coefficients.

**A) Sparsity-optimized** minimal images       **B) Sparsity-Logit** pareto front

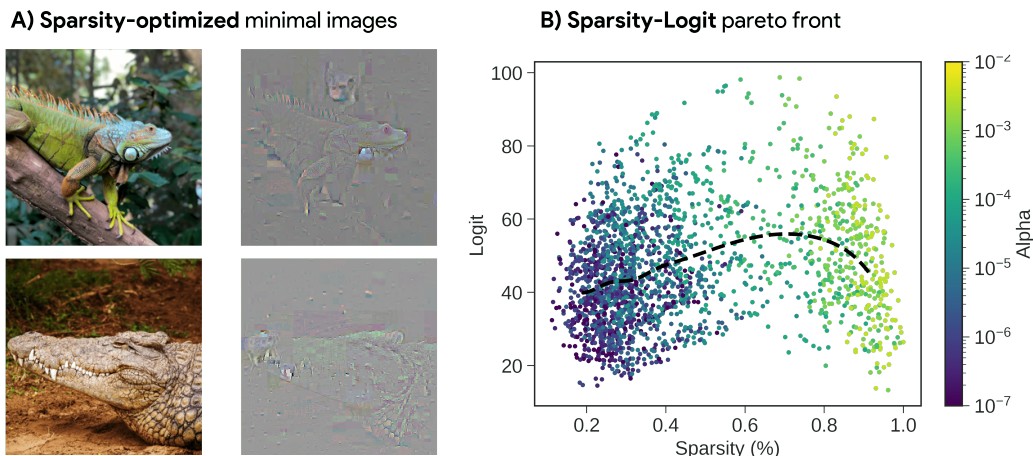

*Figure 11.* **A) Sparsity-optimized minimal images.** We revisit meaningful perturbation by optimizing the sparsity of the wavelet transform using masking, instead of optimizing the mask in pixel space. The displayed examples show that the resulting minimal images reveal the model's reliance on textures. **B) Sparsity Pareto front.** As $\alpha$ increases, the sparsity of the wavelet coefficients increases (x-axis), but beyond a certain point, too much information is lost and the logit score drops to zero. However, we observe that many components can be removed before adversely affecting the model. Results are averaged across 1,000 images optimized for 500 steps and for $\alpha$ ranging in $[0, 100]$ for each image.

Traditional meaningful perturbation methods (Fong & Vedaldi, 2017) focus on spatial localization, identifying clusters of pixels that answer the question of *where* the important features are located. However, this spatial emphasis alone provides a limited understanding of the underlying data structure. In contrast, by operating in the wavelet domain, our method captures both the *what* – the relevant scales – and the *where* – their spatial locations. This dual information enriches the explanation by revealing the location and the nature of the features influencing the model's decision. These results also show that we qualitatively recover the results from Kolek et al. (2023).

Figure 11 illustrates that minimal images derived using WAM recover the texture bias of the vanilla ResNet models trained on ImageNet, highlighted by Geirhos et al. (2019). The examples demonstrate how the model relies heavily on texture information, which is effectively isolated through our wavelet-domain optimization.

**Effect of the regularizer on the sparsity of the images.** Figure 12 illustrates the effect of the parameter $\alpha$ on the sparsity of the minimal images. We can see that the stronger $\alpha$, the sparser the image, but at the expense of a higher logit value. We can see that the first components of the image that disappear are the background, then the colors, and eventually the shape of the target class.

$\alpha = 0.01$      $\alpha = 0.1$      $\alpha = 1.0$      $\alpha = 10.0$      $\alpha = 100.0$

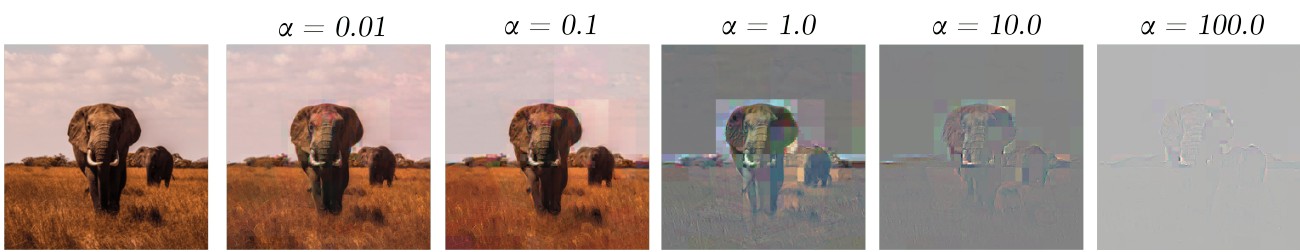

*Figure 12.* **Effect of varying values of $\alpha$ on the sparsity of the minimal images.**

**Minimal images and applications.** Figure 13 presents additional examples of minimal images. We can see that the color information does not appear as important for maximizing the model's prediction. On the other hand, the texture and edge information are essential. It would be interesting to replicate this method on a shape-biased model such as those proposed by Chen et al. (2020) or Geirhos et al. (2019) to see whether the behavior remains the same or not.

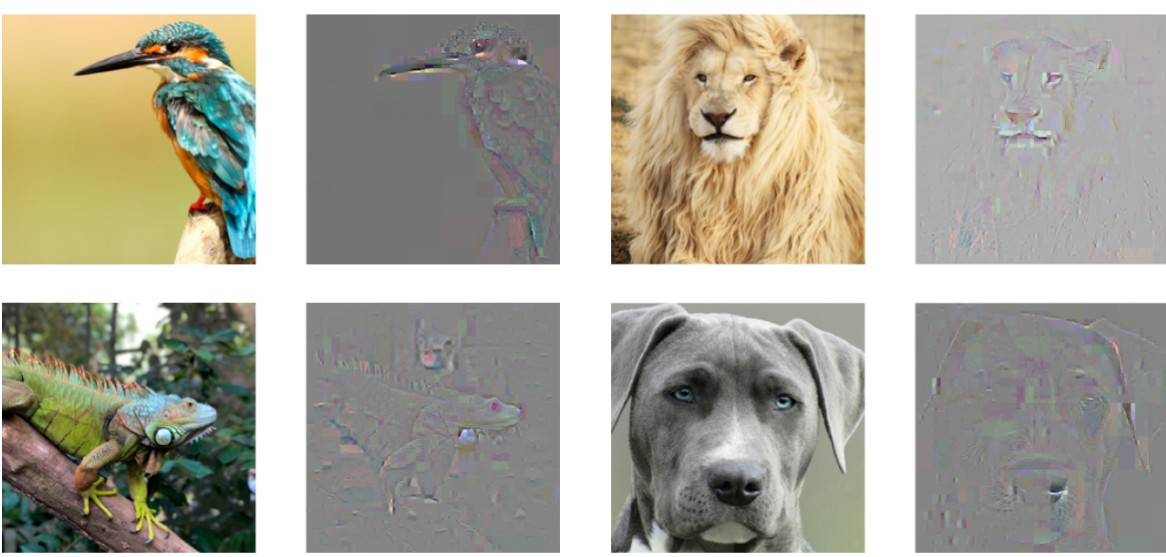

*Figure 13.* **Additional examples of minimal images.**

### D.2. Multi-class classification

**Evaluation in a multi-class setting.** We evaluate WAM in a multi-class setting to assert whether our method is still informative when an image depicts different objects. To evaluate our method, we rely on the Pointing Game (Zhang et al., 2018) benchmark. The Pointing Game assesses the spatial precision of saliency or attribution maps. For each image, the location of the maximum activation in the saliency map (the "point") is compared against the ground-truth object location. If the point lies within the ground-truth bounding box (or segmentation mask), it is considered a "hit"; otherwise, it is a "miss".

The final accuracy is computed as:

$$\text{Pointing Game Accuracy} = \frac{\#\text{Hits}}{\#\text{Hits} + \#\text{Misses}}.$$

This metric focuses on whether the most confident localization prediction corresponds to the true object location, without requiring full object delineation. We evaluate WAM alongside competing methods (GradCAM, GradCAM++, and SmoothGrad) on the PASCAL VOC 2012 test set (Everingham et al., 2010), a widely used benchmark for visual recognition tasks. The dataset contains 20 object classes and 1456 test images. Each image is annotated with object bounding boxes and class labels, making it suitable for evaluating weakly-supervised localization via the Pointing Game.

Table 8 presents the results: we can see that WAM outperforms the competing baselines at the Pointing Game, thereby showing its ability to identify the relevant spatial parts of the input image for a given class. This further backs the fact that the wavelet domain is informative in multi-category images and suggests that WAM can be adapted to object detectors.

**Visualizations.** Once shown that WAM can effectively distinguish objects in multi-category settings, we can analyze which regions are highlighted as important by WAM on such images. In Figure 14, we observe that the WAM successfully highlights "cat-related coefficients," as the cat's nose and its corresponding regions in the wavelet domain are more active in the rightmost image. Interestingly, although less prominent, parts of the dog's face also remain significant. This suggests

*Table 8.* **Pointing game experiment results**. Evaluation carried out on the test set of the PASCAL VOC 2012 test split. Best results are **bolded** and second best underlined.

| Model | ResNet | EfficientNet | ConvNext | VGG |
|---|---|---|---|---|
| GradCAM | 0.476 | 0.454 | 0.528 | 0.437 |
| GradCAM++ | 0.476 | 0.454 | 0.528 | 0.437 |
| SmoothGrad | 0.429 | 0.432 | 0.430 | 0.430 |
| $WAM_{SG}$ (ours) | **0.606** | 0.584 | 0.584 | 0.538 |
| $WAM_{IG}$ (ours) | 0.604 | **0.601** | **0.632** | **0.586** |

that while the spatial separation between the cat and dog is clear in the original image, the distinction becomes less apparent in the wavelet domain.

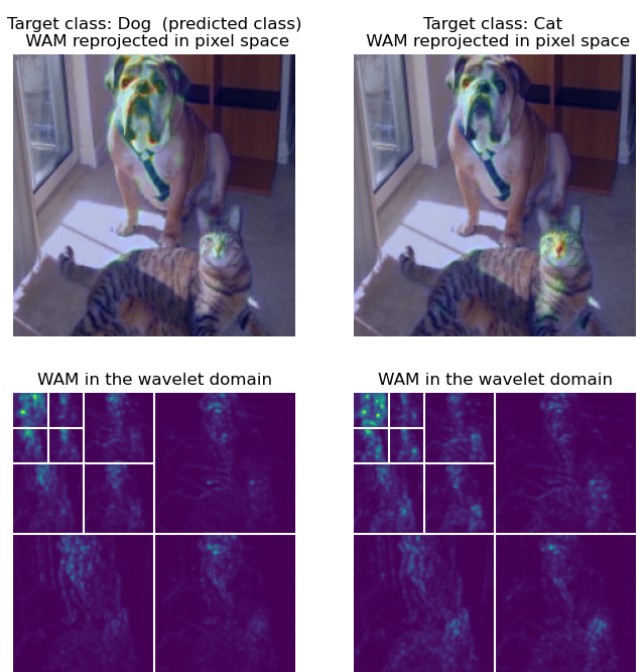

*Figure 14.* **Illustration of multi-category images.** Example on an image depicting a cat and a dog to highlight that the WAM points towards the regions corresponding to the target label (Dogs on the right, Cats on the left).

## D.3. Overlap experiment in audio classification

Figure 15 illustrates that WAM is able to filter relevant parts corrupted or mixed audio signals. In addition, it highlights the key part of the target signal without requiring any training. Figure 15 qualitatively illustrates application of WAM for audio signals. Herein, we perform an overlap experiment to mix a corrupting audio with a target audio to form the input audio. The model's prediction does not alter after introducing the corruption, and thus, the model is expected to still rely on parts of input audio coming from the target audio for its decision. The interpretation audio in Figure 15 generated using top wavelet coefficients provides insights into the decision process and supports this hypothesis. In particular, it almost entirely filters out the corruption audio, and without requiring any training, it also clearly emphasizes key parts of target audio.

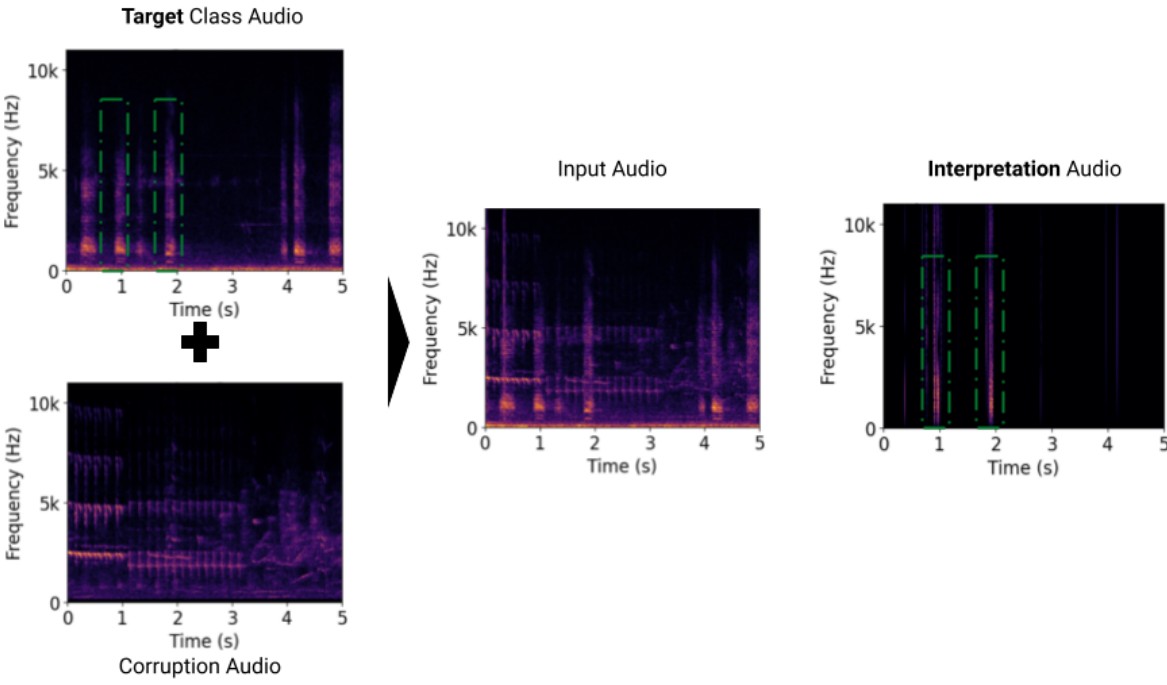

*Figure 15.* **Qualitative illustration of WAM for audio via an Overlap experiment.** The audio of the target class ("Crow") is mixed with a corrupting audio ("Chirping birds") to form the input to the classifier. Interpretation audio reconstructed with important wavelet coefficients virtually eliminates signal from the corrupting audio, and also clearly emphasizes parts of the target class audio (indicated with green boxes).

## D.4. Scales meet semantics: the case of remote sensing applications

The indexation in scales of the wavelet coefficients finds a natural use case in remote sensing applications (Kasmi et al., 2023b), where scales correspond to actual physical objects, as the pixels correspond to distances on the ground. Therefore, given an aerial image whose ground sampling distance is 20 cm/pixel, the 1-2 pixel scale (the finest details), corresponds to details that have a size comprised between 20 to 40 cm.

On Figure 16, we consider a case where a classification model is trained on a source domain and deployed on a target domain, thus mimicking situations where a model is used for regular updates. We can see that the OOD image is slightly different from the source image, in the sense that it is less noisy. While one would expect the model's prediction not to change, it turns out that the photovoltaic (PV) panel is no longer recognized on the rightmost image. Attribution in the pixel domain only is not very informative as to why this happens. Turning into the wavelet domain, we can better grasp why the model no longer detects the PV panel. The background noise, located mostly in the finest scales, is no longer present on the OOD image, and thus the PV panel is no longer recognized. Kasmi et al. (2025) discussed how wavelet-based attribution methods help improve the reliability of classifiers deployed in such operational settings.

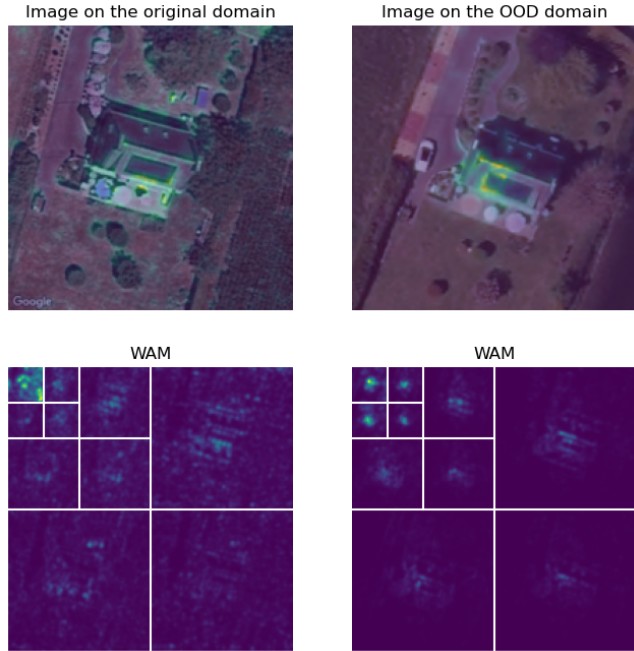

*Figure 16.* **Application case of WAM to remote sensing.** While both images depict the same photovoltaic panel, a model no longer detects the panel when evaluated on images coming from a different distribution ("OOD" or "out-of-distribution" image, compared to images from the source distribution). Attribution in the wavelet domain thanks to the WAM gives hints as to why the PV system is no longer recognized: on the image of the original domain, which is more noisy, the model relies on noise at the finest scales that are no longer present on the OOD image. Thus suggesting that the representation learned by the model lacks robustness. Based on Kasmi et al. (2023b)

