# OpenReview forum: "One Wave To Explain Them All: A Unifying Perspective On Feature Attribution"
_ICML.cc/2025/Conference — ICML 2025 poster_

### Official Review · Reviewer_UvKx · 2025-03-07

**Overall Recommendation:** 2

**Summary:**

The paper explores feature attribution by determining the importance of individual wavelets in prediction tasks. Unlike traditional vision-based attribution methods that assess pixel importance, this approach evaluates how wavelets contribute to model predictions. The key idea is to compute gradients with respect to wavelets to measure their significance. Extensive experiments are conducted to validate the effectiveness of the proposed method.

**Claims And Evidence:**

The paper claims to introduce a new method that improves  interpretability in feature attribution by identifying the importance of individual wavelets. However, while the method assigns importance scores to wavelets, it is unclear how this directly improves interpretability. Wavelets are not inherently interpretable features, and the paper does not provide sufficient justification for why identifying important wavelets leads to a more interpretable representation.

**Essential References Not Discussed:**

The paper's presentation may give the impression that XAI has been primarily focused on vision-based applications, as stated in “While XAI has been predominantly applied in image classification, it is also extending into other fields, such as audio and volume classification.” However, XAI has also been widely explored in other domains, such as NLP, where numerous explainability methods have been developed and studied. Citing relevant works from NLP and other areas could provide a more comprehensive view of the broader landscape of XAI research (e.g., Towards Faithful Model Explanation in NLP: A Survey, A Comparative Study of Faithfulness Metrics for Model Interpretability Methods, Faithfulness Tests for Natural Language Explanations)

**Experimental Designs Or Analyses:**

--

**Methods And Evaluation Criteria:**

The authors select one dataset per modality out of the three considered, which is a reasonable choice given the scope of their study. Additionally, they assess faithfulness using the Insertion and Deletion metrics. While these provide valuable insights, it is worth noting that faithfulness can be defined in multiple ways, and alternative definitions may offer complementary perspectives. For example, in the context of NLP, the survey "Towards Faithful Model Explanation in NLP: A Survey" discusses various definitions of faithfulness, many of which share common ideas with vision-based approaches.

**Other Comments Or Suggestions:**

The title suggests a unifying perspective, but it is unclear what exactly is being unified. At first glance, it resembles the SHAP paper, which unifies different explanation methods for various attribution techniques. Clarifying what is being unified—whether it's explanation methods, domains, or something else—would provide better clarity for the reader.

**Other Strengths And Weaknesses:**

**Strength**: The paper presents a nice idea that is innovative and has potential value in the field. The authors have clearly invested significant time and energy into the experimental aspects of the work, which is evident from the detailed experiments.

**Weakness**: A primary weakness of the paper is that wavelets, as used in the proposed method, are not inherently interpretable features. This raises concerns about the overall interpretability of the method, as the use of wavelets may undermine the transparency of the approach. Additionally, the presentation of the algorithm could be improved. Specifically, when the authors mention "directly evaluate ∂fc(W−1(z))/∂z” it is unclear how this is achieved. Providing more clarity on the exact steps of this evaluation would significantly improve the paper's presentation and make the methodology more accessible to readers.

**Questions For Authors:**

--

**Relation To Broader Scientific Literature:**

The paper presents an approach positioned within explainability research, contributing to the broader literature by examining the importance of different wavelets. Prior work in explainability has typically emphasized methods that rely on interpretable features, ensuring that the extracted insights align with human understanding. In contrast, this paper leverages non-interpretable features, which, while valuable in their own right, may not directly advance explainability in its conventional sense. Given this, it may be beneficial to consider situating the paper within a broader methodological context—perhaps as an approach that improves model performance or provides alternative forms of analysis—rather than as a direct contribution to explainability. Such a reframing could more accurately reflect its impact within the field.

**Theoretical Claims:**

There are no theoretical claims in the paper

---

> ### Author Rebuttal · Authors · 2025-03-31
>
> We first would like to thank reviewer UvKx for the reviews on our work and underlining the potential of our idea, regarded as innovative. We would like to address the comments raised by the reviewer on our work. These comments will be taken into account in our work and will help us improve the quality and the clarity of our work.
>
> ### Interpretability of wavelet coefficients
>
> The reviewer challenged our assertion that wavelet coefficients are interpretable coefficients and stated that we did not provide enough justification. We kindly refer the reviewer to **section 2.2, page 3, where we explain why wavelets are interpretable features for feature attribution**. We add the wavelet transform of a signal provides a more interpretable representation of this signal than raw pixels or Fourier transforms by **capturing spatial and frequency information**. Fourier transforms only encode frequency content whereas wavelets preserve localization, making them particularly effective for analyzing structured data like audio and images. In audio, for instance, wavelets isolate transient components corresponding to phonemes, yielding a more intuitive representation of meaningful patterns. Wavelets **extract high-level features that better convey the overall structure of a signal**, making them a more interpretable alternative to the original data representation. **For images, wavelet coefficients can be intuitively understood by viewing them as capturing details at different scales**, like how our vision works when we look at an image from different distances. **Wavelet coefficients align with the definition of interpretability by Kim et al. (2016, [1]), as the decomposition into different scales enables users to predict how the method will emphasize on specific features (e.g. edges or gridded patterns on images)**. We would like to thank the reviewer for pointing this out and will further enrich section 2.2. with the provided expanation.
>
> ### On the scope and positioning of the paper and our claim on unification
>
> The reviewer stated that our paper could be situated amond approaches that improve model performance or alternative forms of analysis. While this perspective is interesting, as our work indeed bridges the gap between explainability and model robustness for instance, **we believe that our work aligns well with explainability and feature attribution as we essentially propose to attribute an importance score to input features, taken as the wavelet coefficients of the signal of interest rather than its pixels** as in traditional feature attribution methods.
>
> The reviewer mentioned that where our identification lies was unclear, a concern shared by R1. We thank the reviewer for this remark and acknowledge that indeed, we remain elusive on what we aimed at unifying. **Unlike SHAP, we do not unify attribution methods but rather unify the domain in which attribution is carried out in a single input representation, the wavelet transform of this signal**. We edited our manuscript to reflect the changes.
>
> ### Clarifications on the evaluation of $\partial f_c(W^{-1}(z))/\partial z$
>
> The reviewer mentioned that it was unclear how the computation of $\partial f_c(W^{-1}(z))/\partial z$ was achieved. In Pytorch, we **require the gradients on the wavelet transform of the signal $z$** and reconstruct the original image from its wavelet coefficients, $x=W^{-1}(z)$. We **carry out a forward and compute the derivative of the prediction with respect to the wavelet coefficients**. We then re-express the signal into its wavelet transform to retrieve the gradients of the model with respect to the wavelet coefficients of the input signal. **The novelty lies in expanding existing attribution methods (SmoothGrad and IntegratedGrad) into the wavelet domain**.
>
> ### Discussion of the faithfulness metric & additional references
>
> The reviewer highlighted an additional definition of the faithfulness, as discussed in "Towards Faithful Model Explanation in NLP: A Survey". We thank the reviewer for this reference and will add it to our manuscript. **Regarding our definition of the faithfulness, we acknowledge that there are several definitions of this notion**. For this work, we chose the definition of Muzellec et al (2023), but as **we believe that one metric is not sufficient to reflect the behavior of a method, we extensively evaluated our method with alternative metrics in the supplementary materials, section B.2, p.16**.
>
> The reviewer underlines that we do not discuss references from NLP. We would like to recall that **since our method cannot be theoretically applied to text data, we did not considered works in this field (see Introduction, second column, l47-48), although we acknowledge that there have been many works in the field of XAI for NLP**. We thank the reviewer for the reference and will add it to our manuscript.
>
> - [1] Kim et al, 2016. “Examples Are Not Enough, Learn to Criticize! Criticism for Interpretability.” NIPS’16

---

> > ### Comment · Reviewer_UvKx · 2025-04-03
> >
> > The authors clearly took the time to write a detailed rebuttal, which I appreciate.
> >
> > However, even after rereading Section 2.2, I still do not understand why wavelets are considered interpretable features. Since the authors rely on a feature importance method, it is crucial that the features themselves are interpretable. In tabular data, interpretability typically refers to features like "age" or "height"—concepts that are inherently understandable to humans. How do wavelets fit this criterion? The authors argue that wavelets are more interpretable than the Fourier transform, but being "more interpretable" than Fourier does not necessarily make them interpretable in an absolute sense.

---

> > > ### Author Response · Authors · 2025-04-04
> > >
> > > We appreciate the prompt response and rebuttal acknowledgment from the reviewer.
> > >
> > > Individual input features for tabular data are indeed highly interpretable. This **high interpretability however is not available for input features of image/audio/3D data (pixels/audio samples/voxels)**. For such domains concept based/prototypical explanations fill this gap, where individual units of interpretation (concepts/prototypes) are regarded as human understandable. **We agree that wavelet coefficients are not as understandable as concepts/prototypes.**
> > >
> > > However, concept or prototypical explanations are extracted through internal representations of the network. They require information about network architecture and also explicit access to internal layers, which is not possible in many cases (eg. for proprietary models accessible only via APIs). This is why post-hoc attribution methods and saliency visualization are still important as they are the only tools to offer insights in such cases. **Our method and its comparison is against post hoc attribution methods for the aforementioned data domains, which do not require internal model information as prototypical or concepts-based approaches do**. Among these methods, as we argue in the rebuttal (to you and reviewer KCeN) and paper (Sec 2.2), **wavelets provide a more suitable and interpretable representation than raw input features, super-pixels or Fourier coefficients to perform attribution**. Moreover, they also provide a clean pathway to unify attribution methods for multiple modalities, i.e. image/audio/3D data.
> > >
> > > Here is an application that we hope will make the wavelet coefficients more intuitive: the textures isolated by wavelet coefficients can correspond to features in medical images, such as glaucoma [1]. The advantage of classifying glaucoma images using wavelet-based descriptors is that these descriptors are fixed and defined in a closed form, as opposed to features from CNN models, which can be hard to decipher. Our approach leverages the expressivity of wavelets to explain modern—and more accurate—classification models.
> > >
> > > In summary, while we acknowledge the validity of your underlying point, we believe that WAM should be evaluated in comparison to post-hoc attribution methods for image/audio/3D data, where interpretable concepts either do not exist or require access to the model's inner layers. In contrast, the wavelet transform can capture and disentangle **textures, edges, and other patterns that correspond to intuitive attributes that vary depending on the context** (e.g., glaucoma, as mentioned earlier, or fields and roads in remote sensing images, veins in leaf images, etc.).
> > >
> > > - [1] Dua, S., Acharya, U. R., Chowriappa, P., & Sree, S. V. (2011). Wavelet-based energy features for glaucomatous image classification. Ieee transactions on information technology in biomedicine, 16(1), 80-87.

---

### Official Review · Reviewer_EDJ8 · 2025-03-12

**Overall Recommendation:** 3

**Summary:**

This paper proposes an explanation method for DNN by using wavelet coefficients as features for attribution instead of image features. The proposed WAM can be adopted across diverse modalities, including audio, images, and volumes. It unifies and extends existing methods, SmoothGrad and Integrated Gradients, within the wavelet domain by transporting the gradient w.r.t. the input to the transformed wavelet domain.

## update after rebuttal
Thank the authors for the detailed rebuttal. My concerns are basically addressed. For the results of localization evaluation (Point Game) and analysis of multi-class cases, the authors haven't updated the rebuttal yet. Still, I strongly recommend adding these results to the final version of the paper. I will keep my rating.

**Claims And Evidence:**

- Using wavelets to decompose image information is a novel view for interpreting the model’s decision for obtaining what information the model has seen when making the prediction. The methodology seems reasonable and claimed clearly.
- The key part of the proposed wavelet method is transporting the gradient w.r.t. original input to the gradient w.r.t. the transformed input (wavelet domain). This limits that the proposed method cannot adopt the intermediate features generated from the model, which are used more by other gradient-based explanation methods [A, B, C]

[A] Selvaraju R R, Cogswell M, Das A, et al. Grad-cam: Visual explanations from deep networks via gradient-based localization[C]//Proceedings of the IEEE international conference on computer vision. 2017: 618-626.

[B] Jiang P T, Zhang C B, Hou Q, et al. Layercam: Exploring hierarchical class activation maps for localization[J]. IEEE Transactions on Image Processing, 2021, 30: 5875-5888.

[C] Zhao C, Chan A B. ODAM: Gradient-based Instance-Specific Visual Explanations for Object Detection[C]//ICLR. 2023.

**Essential References Not Discussed:**

N/A

**Experimental Designs Or Analyses:**

- About the visualization results in A.3, the distinction between dog and cat as target class is less apparent in the wavelet domain. Does that demonstrate that when there is a multi-category in the image, the model cannot be interpreted well in the wavelet domain? Should give more examples and discussion on this kind of case.

**Methods And Evaluation Criteria:**

- For evaluating explanation methods for image inputs, except faithfulness evaluation, it’s also better to provide localization evaluation, Pointing Game [D], and visual explanation accuracy [E].
[D] Zhang J, Bargal S A, Lin Z, et al. Top-down neural attention by excitation backprop[J]. International Journal of Computer Vision, 2018, 126(10): 1084-1102.
[E] Oramas J, Wang K, Tuytelaars T. Visual explanation by interpretation: Improving visual feedback capabilities of deep neural networks[J]. arXiv preprint arXiv:1712.06302, 2017.

**Other Comments Or Suggestions:**

- duplicated reference for Grad-CAM (Page 11 Line 593-604)

**Other Strengths And Weaknesses:**

N/A

**Questions For Authors:**

The questions mentioned above and

Is that possible
1) adopting wavelet in the feature layer;
2) wavelet-based explanation method for interpreting Transformer;
3) wavelet-based explanation method for other task models like object detectors and VLMs?

**Relation To Broader Scientific Literature:**

This paper proposes a way to interpret the model with inputs of audio and volumes. Almost all previous explanation methods are designed for models with image inputs.

**Theoretical Claims:**

Seems no problem.

---

> ### Author Rebuttal · Authors · 2025-03-31
>
> We thank reviewer EDJ8 for the review and the comments on our work. We also thank the reviewer for pointing the duplicate reference, which we have corrected.
>
> ### Regarding the evaluation criteria
>
> The Reviewer said to include localization evaluation, Pointing Game, and visual explanation accuracy alongside faithfulness evaluation for assessing explanation methods for image inputs. We thank the reviewer for providing additional benchmarks. **We kindly refer the reviewer to the supplementary material, as we have evaluated our method accross a wide range of common metrics from the existing literature**. To complement these results, **we are currently implementing the Pointing Game and Visual Explanation Accurary, as suggested by the reviewer, and will update the rebuttal if we get results**.
>
> ### Regarding the multiclass classification (Cats and Dogs examples)
>
> The reviewer outlined the fact that from appendix A3, the distinction between dog and cat as target classes is less apparent in the wavelet domain and suggested providing more examples and discussion on multi-category interpretation challenges. **To be more conclusive regarding the assertion ''Does that demonstrate that when there is a multi-category in the image, the model cannot be interpreted well in the wavelet domain'', we are gathering more examples**. Localization benchmarks should also give us information on the behavior in multi-category settings. **We try our best to get the results and will update the rebuttal**.
>
> ### Questions
> *1. adopting wavelet in the feature layer;*
>
> This is an interesting question and was actually planned for future work. **To the best of our knowledge, no work has adopted wavelets in the feature layers**. Like [A,B] and contrary to the statement of the reviewer, the wavelet decomposition does not prevent us from using the intermediate layers of the model to compute the explanations. **In principle by applying wavelet decomposition to intermediate feature maps, we could obtain a "multiscale interpretation" of feature importance**.
>
> *2. wavelet-based explanation method for interpreting Transformer;*
>
> If the reviewer refers to the interpretation of the attention mechanism using wavelets, it is an interesting suggestion but beyond the scope of this work. **If it is meant applying WAM to Transformer-based architectures, then we kindly refer the reviewer to the supplementary materials B2 were we apply the WAM to a wide variety of topologies (ViT, ConvNext, 3D Former)**. Results remain the same no matter the choice of the classification model.
>
> *3. wavelet-based explanation method for other task models like object detectors and VLMs*
>
> These two comments are a very good suggestion and we would like to thank ther reviewer for these remarks. We remained focused on the classification task but **in principle, our method could be expanded to object detectors. The way to follow would be to expand ODAM to the wavelet domain**. Regarding VLMs, since our method unifies the domain for feature attribution it makes sense to look for interpreting multimodal models. However **VLMs handle text data, and this modality is unsuitable for the wavelet transform, so attribution in a suitable latent space for text data shoud be explored**. We are currently applying WAM to object detectors and will update the rebuttal if we get results.

---

### Official Review · Reviewer_KCeN · 2025-03-14

**Overall Recommendation:** 3

**Summary:**

Presents a feature attribution method that performs attributions on wavelets derived from input domain. This helps to naturally extend explanations that are outside the image domain, such as an audio input domain. The method essentially constructs a wavelet transform of the input, then applies standard gradient and IG attribution methods to wavelet-domain inputs. Insertion/deletion tests for faithfulness are performed to compare WAM to other popular methods across audio, 3d, and image domains.

## update after rebuttal
After reading the rebuttal, I am confident in my original assessment. I encourage the authors to include analysis of choice of $\Lambda$ and various wavelet choices, which I could not find in the original paper. We also encourage the author to address if wavelets have any absolute edge over all other methods and are, in some analytic sense, better than all other possible methods, or if wavelets just have a comparative edge over some current popular methods. Upon further reflection, we also recommend an edit for readability.

**Claims And Evidence:**

The method, WAM, does appear to have good performance compared to other popular gradient-based attribution methods over a variety of tests.

The paper demonstrates some advantage to using wavelets, i.e. for attributing at different feature scales.

**Essential References Not Discussed:**

None known.

**Experimental Designs Or Analyses:**

Insertion/deletion is an appropriate metric, and at least the visual dataset is standard, to my knowledge.

**Methods And Evaluation Criteria:**

The faithfulness metric is an appropriate method, and the evaluation sets seem appropriate.

**Other Comments Or Suggestions:**

Pg 4, line 209, left: "Varying pixel values provide no information to what is changing on the image." Please expound. Changing on the image? No information to what?

Your paper claims WAM is a unifying perspective of feature attribution. When I read this, I expected the paper to present an analysis that unifies multiple attribution methods under one theory. However, the paper seems to provide a method that is adaptable to multiple input domains. Perhaps some more up-front clarity/renaming would clarify this fact and prevent confusion.

I can see how wavelets is another way to attribute to the input domain, so that attributions now incorporate a sense of scale. While this is an advantage, can you provide any argument as to why wavelets are $\textit{the}$ appropriate terms in which to attribute? What about super-pixels? What about Fourier transforms and attributing in k-space? Is this method more appropriate than those, theoretically or experimentally?

**Other Strengths And Weaknesses:**

See questions and suggestions

**Questions For Authors:**

What wavelet functions were chosen for each experiment? What $\Lambda$ was chosen? Why were they chosen? Sorry, I did not catch this on my read though; perhaps they should be presented more prominently.

How does the faithfulness score and the quality of explanations vary with the choice of wavelet function and choice of $\Lambda$? Is it sensitive to these choices?

A major component of IG is completeness, i.e., the total change in function value at the baseline vs input value is accounted for by the attributions. This gives meaning to the value of the IG output: i.e., the value of an IG attribution is equivalent to function change. Does WAM_IG satisfy completeness?

**Relation To Broader Scientific Literature:**

The paper piggybacks off of the gradient-based attribution literature (Sundararajan IG paper), adapting the method to better handle models for volume and audio input domains, by using wavelets. Thus it contributes to (the small amount of) literature on wavelet based attributions.

**Theoretical Claims:**

I looked over the appropriateness of eq 4.

---

> ### Author Rebuttal · Authors · 2025-03-31
>
> We first would like to thank reviewer KCeN for reviewing our manuscript and for the comments on our work. The points raised by the reviewers will help us improve the quality of our work.
>
> **Question** *Pg 4, line 209, left: [...] Please expound / I can see how wavelets is another way to attribute to the input domain, so that attributions now incorporate a sense of scale. [...] can you provide any argument as to why wavelets are appropriate terms in which to attribute? What about super-pixels? What about Fourier transforms and attributing in k-space? Is this method more appropriate than those, theoretically or experimentally?*
>
> Our formulation "varying pixel values provide no information to wht is changing on the image" is unclear and we modified it for "moving from one pixel to the next is only a shift in the spatial domain and does not capture relationships between scales or frequencies". This sentence explains why in our view, the pixel domain is insufficient to interpret the decision of a model. **The wavelet domain captures both the spatial component (as done by pixels, or superpixels) and the spectral component, as done by the Fourier transform**. Either one of these approaches is unsuitable for attribution (see [1,2] for Fourier), as **for attribution a spatial or temporal dimension is required and a spectral dimension desirable** to assess what the model sees at a given location.
>
> **Question** *What wavelet functions were chosen for each experiment? What was chosen? Why were they chosen? [...] perhaps they should be presented more prominently. / How does the faithfulness score and the quality of explanations vary with the choice of wavelet function and choice of $\Lambda$ ? Is it sensitive to these choices?*
>
> **The quality and faithfulness of explanations does not change with the choice of the mother wavelet**. the quality and faithfulness of the explanation stem from the multi-scale decomposition, which is a property of the $\Lambda$, irrelevant of the choice of the mother wavelet.  The choice of the mother wavelet determines how the input signal is decomposed, influencing whether finer details or broader structures are emphasized in the wavelet coefficients By default, we considered the Haar wavelet, due to its semantic properties but also because it is fast to compute.
>
> **Additional experiments verified that the choice of the wavelet does not change the quantitative results**. We evaluated $WAM_{IG}$ on a ResNet 50 model using Daubechies and Bior wavelets. We updated the supplementary materials in to state more clearly our choice of wavelet and discuss what wavelets can be chosen.
>
> On the other hand, **the choice of $\Lambda$, remained fixed and we did not explore beyond the dyadic transform because the multi-scale property of our decomposition fundamentally relies on its dyadic structure**. Using non-dyadic decompositions could disrupt the natural hierarchy of scales, leading to a loss of spatial localization and a less structured frequency representation.
>
> **Question** *Your paper claims WAM is a unifying perspective of feature attribution. [...]. However, the paper seems to provide a method that is adaptable to multiple input domains. Perhaps some more up-front clarity/renaming would clarify this fact and prevent confusion.*
>
> We acknowlege that we remained elusive on what we aimed at unifying, a concern shared by R3. We will edit our manuscript to explicitely state what we aim to unify in this work.
>
> **Question** *A major component of IG is completeness. [...] Does WAM_IG satisfy completeness?*
>
> For completeness, only **showing that inverse wavelet transform keeps things differentiable and that the initial and final points are same is enough (see [3], Prop 1)** to show that $WAM_{IG}$ satisfies completeness.
>
> Formally, IG satisfies completeness if $F$ is differentiable almost everywhere and
> $$
> \sum_{i=1}^n IntegratedGrad_i(x) = F(x) - F(x_0)
> $$
>
> Where $x_0$ is a baseline black image such that $F(x_0)\approx 0$. To ensure that $WAM_{IG}$ satisfies the condition for completeness, we have to ensure that (1) $F(W^{-1}(z))$ is differentiable almost everywhere and that (2) we can set $z_0$ such that such that $W^{-1}(z_0)$ is a black image. **(1) depends on the choice of the mother wavelet $\psi$** (we need the mother wavelet to be smooth) and **(2) is obtained by setting the wavelet coefficients to 0**, so **$WAM_{IG}$ satisfies completeness if $\psi$ is smooth**, e.g. with Daubechies but not with Haar wavelets.
>
> - [1] Yin et al (2019). A fourier perspective on model robustness in computer vision. NeurIPS
> - [2] Chen et al (2022). Rethinking and improving robustness of convolutional neural networks: a shapley value-based approach in frequency domain. NeurIPS
> - [3] Sundararajan et al (2017) Axiomatic attribution for deep networks. ICML

---

### Decision · Program_Chairs · 2025-05-01

**Decision:**

Accept (poster)

**Comment:**

The paper was reviewed by 3 experts, with mixed initial reviews 233. The major issues raised were:

1) What is the "unifying perspective" of feature attribution? [KCeN, UvKx]
2) Why are wavelets the most appropriate to attribute compared to super-pixels, Fourier transform, etc.? [KCeN]
3) Which wavelet functions were selected and why? How does the evaluation/quality change with choice of wavelet?
4) Does WAM_IG satisfy completeness? [KCeN]
5) Method cannot be adopted to the intermediate features generated from a model (e.g., as with Grad-CAM methods). [EDJ8]
6) could use other metrics for image explanations, such as pointing game or visual explanation accuracy  [EDJ8]
7) regarding images in A.3, can multi-category images be interpret well in the wavelet domain? [EDJ8]
8) could the same method be used for a feature layer, transformers, or more complex models for other tasks? [EDJ8]
9) wavelets are not inherently interpretable features, how does identifying wavelets lead to more interpretable representations? [UvKx]
10) experiments could use other faithfulness metrics [UvKx]
11) contribution should be reframed since it uses non-interpretable features, which may not directly advance explainability in the conventional sense [UvKx]
12) presentation could be improved [UvKx, KCeN]

The authors wrote a response. After the response, KCeN was satisfied, maintained rating, and comments that the paper could be further edited. In particular:
- "Well grounded progress in attributions must provide, in part, some analysis showing the unique advantages of the method or its well-suitedness to a problem domain. This is very common in the field; see axiomatic approaches. This method does not provide much in that regard; the justifying paragraph seems a bit undeveloped and perfunctory. On the other hand, many methods have been introduced that lacked such analysis and yet were interesting and enlightening to read. The lack of such analysis meant the method lacked theoretical maturity when it was introduced, but the idea and experiments may have been of practical interest or inspirational for later work. That being said, doing some sort of wavelet explainability method seems like a thing that should be done, and the analysis here has some merit. However, the lack of theoretical development (theorems, in-depth discussion of advantages) significantly limits the impact of the paper. Of course, I would like the paper that first applies wavelets to attributions to have more of a thorough and enlightening review of what a wavelet is, and a better discussion of the advantages."

On the other hand, UVkX was still not convinced that wavelets are interpretable features.

Regarding KCeN's concern, the AC agreed that it would have been nice to present some analysis about how wavelets could be better than others (superpixels, etc). Some analysis probably exists in the literature, e.g., wavelets are commonly used for image/sound compression, and are good at representing transient signals (common for natural images and audio). Perhaps there is also work showing that wavelets are more attuned to the human visual system (via Gabor filters) than super-pixels.

Regarding UvKx's concern about interpretability, it can be noted that wavelets are compact basis functions of different spatial locations and frequency component, so they could be interpreted as spatially-localized frequency content. The analytical form of each basis can be derived mathematically. Gabor filters are an instance of Wavelets, and Gabor filters are considered as interpretable low-level features (orientation- and scale-selective features).  Thus, the AC thinks that wavelets are also interpretable low-level features. The authors could have shown some examples of the wavelets they are using to make this clearer, as well as noting that they are not high-level interpretable features (concepts).

All reviewers kept their ratings after the response. The AC agreed with the two positive reviewers. The method is interesting and novel, and experiment results are compelling.  The idea of using wavelets is likely to inspire other follow-up works.  Thus, the AC recommends accept. The authors should revise the paper according to the comments, taking particular care in introducing wavelets and properly motivating them as interpretable low-level features.